# From EduVisBench to EduVisAgent: A Benchmark and Multi-Agent Framework for Reasoning-Driven Pedagogical Visualization

**Haonian Ji**[1*]**, Shi Qiu**[1*]**, Siyang Xin**[1*]**, Siwei Han**[1*]
**Zhaorun Chen**[2]**, Dake Zhang**[3]**, Hongyi Wang**[3]**, Huaxiu Yao**[1]
[1]**UNC-Chapel Hill,** [2]**University of Chicago,** [3]**Rutgers University**

## Abstract

While foundation models (FMs), such as diffusion models and large vision-language models (LVLMs), have been widely applied in educational contexts, their ability to generate pedagogically effective visual explanations remains limited. Most existing approaches focus primarily on textual reasoning, overlooking the critical role of structured and interpretable visualizations in supporting conceptual understanding. To better assess the visual reasoning capabilities of FMs in educational settings, we introduce EduVisBench, a multi-domain, multi-level benchmark. EduVisBench features diverse STEM problem sets requiring visually grounded solutions, along with a fine-grained evaluation rubric informed by pedagogical theory. Our empirical analysis reveals that existing models frequently struggle with the inherent challenge of decomposing complex reasoning and translating it into visual representations aligned with human cognitive processes. To address these limitations, we propose EduVisAgent, a multi-agent collaborative framework that coordinates specialized agents for instructional planning, reasoning decomposition, metacognitive prompting, and visualization design. Experimental results show that EduVisAgent substantially outperforms all baselines, achieving a 40.2% improvement and delivering more educationally aligned visualizations. EduVisBench and EduVisAgent are available at github.com/aiming-lab/EduVisBench and github.com/aiming-lab/EduVisAgent.

## 1 Introduction

*"To truly teach is not to tell the answer, but to illuminate the path."*

While foundation models (FMs), such as diffusion models and large vision-language models (LVLMs), have been extensively adopted in educational domains (Chu et al., 2025; Wang et al., 2024), including pedagogical agents providing automated classroom assistance and science learning agents offering textual explanations of problem-solving processes (Wu et al., 2023), their applications have predominantly focused on text-based interactions (Wu et al., 2023; Xu et al., 2024). However, in education, especially K-12 settings, creating compelling visualizations is crucial for cognitive comprehension and overall learning effectiveness (Presmeg, 2006). Despite its importance, there is currently limited understanding of how FMs can effectively generate visually grounded

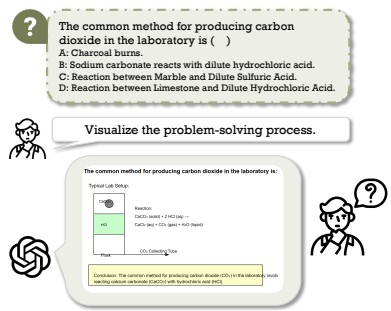

Figure 1: GPT-4o fails to illustrate its problem-solving with high-quality, logical, and explanatory visualization.

elements (e.g., *diagrams*, *interactive education tools*, *illustrative graphics*) to support the pedagogical illustration of problem-solving processes.

---

*Equal contribution. Correspondence to: haonianj@unc.edu, huaxiu@cs.unc.edu

Currently, generating visually grounded elements for pedagogical reasoning poses several challenges: (1) decomposing complex reasoning into representable steps that align closely with human cognitive processes is non-trivial (Yang et al., 2024; Chen et al., 2024d); (2) precisely producing visual aids for each sub-step to optimally support learners is challenging (Hong et al., 2025); and (3) different educational domains require distinct visualization styles and formats, which makes consistent and adequate visual aid delivery difficult (Pandey & Ottley, 2025). This difficulty stems not just from technical rendering challenges, but from the complex task of translating abstract pedagogical concepts into intuitive visual narratives. Addressing these obstacles first requires a picture of how current FMs perform, so that future models can be purpose-built to close the gaps. Consequently, a comprehensive evaluation platform is critical for systematically assessing FMs on visual pedagogical reasoning.

We introduce **EduVisBench**, a multi-domain, multi-level benchmark for evaluating the capacity of FMs to generate pedagogically effective, step-by-step visual reasoning. EduVisBench comprises structured problem sets across diverse domains, each requiring multimodal-centric reasoning and solutions that prioritize visualization principles such as *interpretability*, *cognitive alignment*, and *instructional clarity* to achieve high evaluation scores. We further develop a detailed rubric enabling multidimensional assessment of AI-generated visual outputs across five pedagogical criteria: *contextual relevance*, *visual clarity*, *multimodal coherence*, *reasoning support*, and *interactive engagement*.

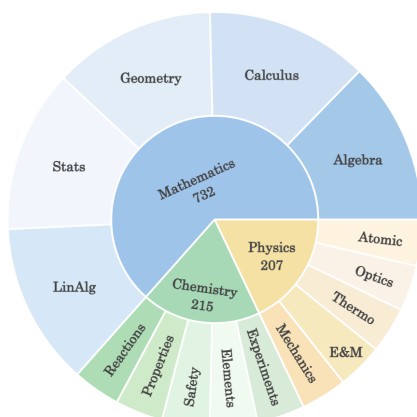

Figure 2: Dataset distribution of EduVis-Bench. Each domain encompasses various sub-domains, collectively covering 15 comprehensive pedagogical scenarios.

Utilizing this benchmark, we conduct extensive evaluations on a variety of FMs and agents. Our findings reveal that although current models achieve predominantly correct step-by-step textual analyses, they frequently fail to generate useful or faithful visualizations, as depicted in Figure 1. Specifically, our systematic analysis highlights recurring challenges including (1) semantic misalignments between textual explanations and visual components, (2) omissions of critical steps within rendered diagrams, and (3) structural inconsistencies in code-based visual outputs, collectively undermining accuracy, clarity, and interactivity. These shortcomings collectively compromise the pedagogical utility of the generated content, often leading to more confusion than clarity for the learner.

To address these limitations, we introduce a multi-agent collaborative framework, **EduVisAgent**, designed to simulate the complete learning journey—from initial problem exposure to deep conceptual understanding. Specifically, a central planning agent orchestrates five specialized expert agents dedicated to *visualization design*, *cognitive scaffolding*, and *metacognitive regulation*. A synthesis module then integrates these expert outputs into interactive, personalized learning webpages tailored specifically to human learners. Experimental results demonstrate that our proposed method EduVis-Agent achieves an average improvement of 40.2% over all baselines. The gains stem from modular specialization across agents and their collaborative integration into a single learning webpage.

## 2 EduVisBench Benchmark

### 2.1 Overview

In this section, we introduce EduVisBench, a benchmark designed to evaluate the capability of models to generate logical and explanatory visualizations for educational purposes. As shown in Figure 2, EduVisBench comprises 1,154 STEM questions across three academic subjects and 15 distinct domains, organized into three levels of difficulty. In addition to assessing accuracy in step-by-step problem solving, EduVisBench places particular emphasis on a model's ability to communicate the reasoning process clearly and visually—helping students understand problems through structured, interpretable visual outputs, as illustrated in Figure 3.

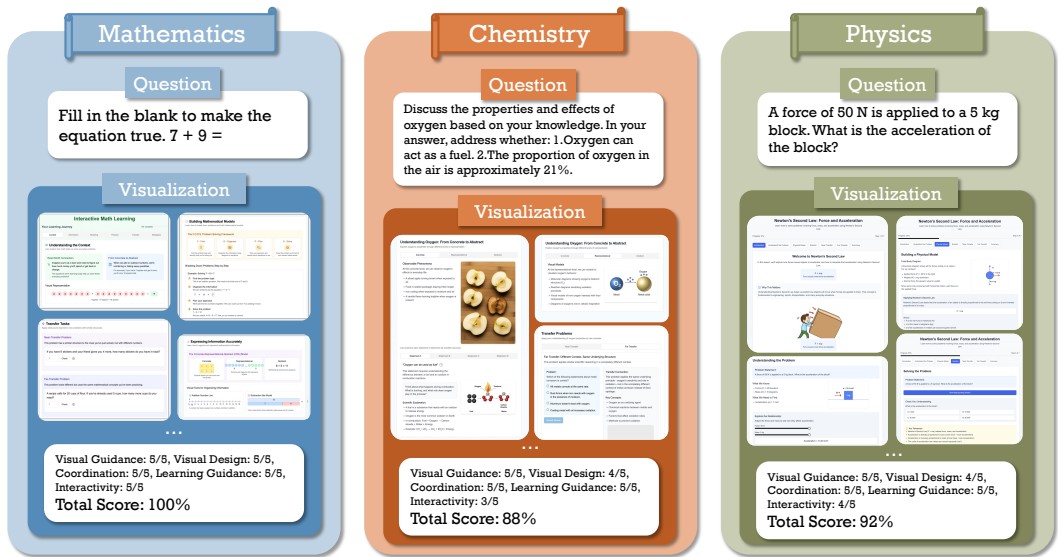

Figure 3: Representative examples from EduVisBench, featuring questions from Maths, Chemistry, and Physics alongside their corresponding high-scoring visual explanations. These interactive visualizations, generated by our multi-agent system EduVisAgent, exemplify well-designed, pedagogically effective outputs for STEM problems.

Specifically, EduVisBench adopts a multimodal setting in which models are provided with both textual and visual inputs and are tasked with producing diverse output formats, including interactive web pages and visual diagrams. Beyond evaluating the correctness of final answers, we introduce a detailed evaluation framework that assesses visualization quality across five dimensions: (1) the logical sequencing of visual elements, (2) the structural richness of the visuals, (3) semantic alignment with the underlying subject matter, (4) the clarity and guidance provided for problem-solving, and (5) the level of interactivity and engagement. In the following subsections, we describe our dataset curation process and the design of the evaluation rubric in detail.

## 2.2 DATASET CURATION

EduVisBench is built from several high-quality public educational resources that we carefully curated, translated, and adapted to support multimodal visualization learning tasks. Specifically, the chemistry questions are sourced from the *C-MHChem-Benchmark* (Zhang et al., 2024), originally presented in Chinese and meticulously translated into English with careful attention to scientific accuracy and terminology. The physics questions are drawn from the *high-school-physics* (Rohith, 2023) dataset, which includes a range of conceptual and quantitative exercises suitable for secondary-level learners. The mathematics component combines easy-level problems from the Illustrative Mathematics curriculum with medium- to hard-level questions selected from the *MATH-500* (Lightman et al., 2023) dataset. Furthermore, each domain encompasses diverse sub-domains, collectively covering 15 comprehensive scenarios, as illustrated in Figure 2. All data sources were standardized into a unified format and consolidated to enable consistent and comprehensive evaluation across subjects.

## 2.3 EVALUATION METRIC

In this subsection, we detail the performance evaluation rubrics in EduVisBench.

**Evaluation Dimensions.** To evaluate the quality of generated visualizations in supporting student understanding, we introduce a scoring metric grounded in five pedagogically motivated dimensions: **(1) Context Visualization**: evaluates how clearly the visualization situates the problem within a relevant context; **(2) Diagram Design**: assesses the clarity, accuracy, and effectiveness of the diagrams used to represent information; **(3) Text–Graphic Integration**: measures the coherence between textual explanations and visual elements, ensuring mutual interaction; **(4) Thought Guidance**: examines the extent to which the visualization supports reasoning processes and highlights critical

thinking steps; **(5) Interactivity**: evaluates whether and how the visualization invites students' engagement, reflection, or active manipulation. Each dimension captures a distinct aspect of effective multimedia learning, with detailed rubrics provided in Appendix A.1 to guide the scoring process.

**Evaluation Protocol.** As shown in Figure 4, models are provided with a visualization prompt together with a question and are asked to generate visual outputs. To enable fair comparison across heterogeneous outputs, we first canonicalize every model result to a raster image prior to scoring. This standardization is a crucial step that ensures all systems are evaluated on a level playing field, independent of their native modality or file format, and prevents format-specific rendering artifacts from biasing the assessment. Visuals produced directly as SVG or PNG are used as-is. Web pages (HTML or Next.js) are rendered in a headless browser and captured as screenshots of the primary view; when lightweight interactivity is present (e.g., buttons, tabs, or toggles), we systematically traverse the reachable states and retain one representative screenshot per state. All resulting images are then evaluated by GPT-4o along five dimensions defined in Appendix A.2 to compute an overall performance score. Each dimension is rated on a 0-5 scale; the ratings are summed (0-25) and, when appropriate, normalized to a percentage to yield the final overall score.

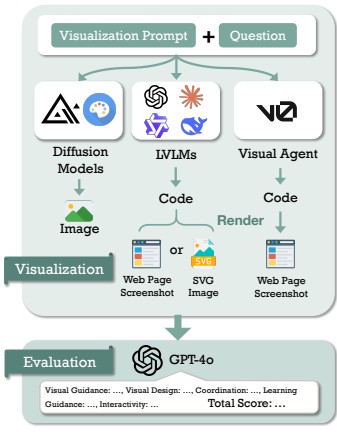

Figure 4: Workflow for evaluation.

# 3 EduVisAgent

Using EduVisBench we systematically evaluate the performance of existing text-to-image models and LVLMs (see detailed results in Table 1 in Section 4). We find that most models perform poorly, with average scores below 50 on a 0-100 scale. This performance gap motivates moving beyond single-model architectures: decomposing complex reasoning into visual representations that support human learning is a genuinely hard problem that no single model currently handles well.

To address these challenges, we propose a multi-agent system, EduVisAgent, inspired by pedagogical theories and designed to emulate the division of labor and collaborative reasoning found in expert instructional design. EduVisAgent consists of five specialized yet interdependent agents: a **Task Planning Agent**, which structures the instructional objective; a **Conceptual Mapping Agent**, which extracts and organizes key information; a **Reasoning Decomposition Agent**, which constructs step-by-step problem-solving logic; a **Metacognitive Reviewer**, which encourages summarization and learner reflection; and a **Visualization Agent**, which generates appropriate visual representations. This design introduces modularity and pedagogical interpretability by embedding distinct instructional roles directly into the agent workflow. The overall operation of EduVisAgent proceeds in two stages: (1) instructional flow construction and (2) collaborative solution generation, as detailed below.

## 3.1 INSTRUCTIONAL FLOW CONSTRUCTION

The first stage of EduVisAgent focuses on formulating a well-structured instructional task based on the original problem. A key challenge lies in analyzing the underlying reasoning structure, identifying implicit logical dependencies, and associating each reasoning step with relevant conceptual knowledge. To address this, we employ the **Task Planning Agent**, which systematically organizes the problem into an instructional format suitable for multimodal visualization. Its main functions include: (1) breaking down the problem into coherent subgoals, (2) clarifying the reasoning expected at each step, (3) aligning each step with domain-specific principles or formulas, and (4) anticipating potential student misconceptions or cognitive needs. This structured formulation provides a pedagogically grounded foundation that guides the downstream agents in generating accurate, educationally grounded visual explanations.

## 3.2 COLLABORATIVE SOLUTION GENERATION

In this stage, EduVisAgent executes the instructional task constructed by sequentially activating a set of specialized agents, each responsible for completing a specific aspect of the task. As shown

in Figure 5, these agents operate in a coordinated manner to enhance the coherence of instructional logic, improve the clarity of visual representation, and ensure alignment with educational objectives. Specifically,we detail each agent as follows.

**Conceptual Mapping Agent.** This agent is responsible for extracting and organizing the core components of the input problem. Drawing on the Concrete–Representational–Abstract (CRA) instructional model (Nugroho & Jailani, 2019), it classifies information into three categories: concrete entities, representational elements, and abstract constructs. This structured classification helps bridge the gap between the concrete elements of a problem and the abstract principles required to solve it. This progression from concrete to abstract is particularly valuable for an AI system, as it provides a structured pathway to ground complex concepts in relatable terms before generating symbolic representations.The agent conducts fine-grained categorization and semantic summarization to support downstream visualization modules.

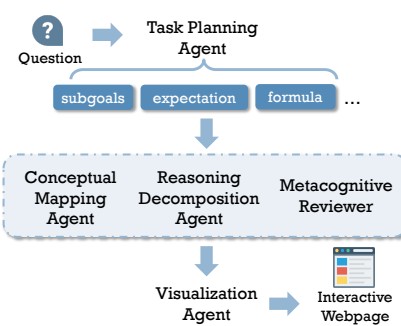

Figure 5: The structure of EduVisAgent.

**Reasoning Decomposition Agent.** This agent decomposes complex problems into manageable subcomponents and provides step-specific instructional guidance. It applies the memory-oriented FOPS strategy (Miller & Cohen, 2020)—find the problem type (e.g., equation solving, conceptual reasoning, commonsense application, or graphical interpretation), organize the structure via equations or diagrams, plan the solution path, and solve the task. Based on the decomposed steps, the agent also identifies critical instructional points that require additional support, especially those that benefit from visual scaffolding or interactive guidance.

**Metacognitive Reviewer.** Grounded in metacognitive theory (Schraw & Moshman, 1995), this agent supports learners in monitoring their comprehension and reasoning processes. It generates reflective prompts that foster self-questioning and self-correction, encouraging learners to evaluate the soundness of their problem-solving approaches.

**Visualization Agent.** This agent is responsible for constructing the "visual guidance" component of the instructional output. Instead of relying on decorative visuals, it emphasizes the use of abstract yet pedagogically effective representations—such as number lines, bar charts, schematic object illustrations, graphic organizers, sketch diagrams, and structured data tables. The agent ensures that each visualization is tightly aligned with the underlying abstract concept being taught. All visuals are rendered using the v0 (Vercel, 2025) system for web-based deployment.

## 4 EXPERIMENTS

This section outlines the experimental setup for benchmarking various foundation models on EduVisBench. We evaluate Diffusion Models, LVLMs, a specialized visualization agent (v0), and our proposed EduVisAgent. Our investigation seeks to address the following key questions: (1) How proficient are existing models at generating high-quality, explanatory visualizations within EduVisBench? (2) Can the proposed EduVisAgent system outperform current models? (3) What distinct performance patterns emerge across different model architectures, academic disciplines, and evaluation dimensions in EduVisBench?

### 4.1 EXPERIMENT SETUP

**Baseline Models.** Our experimental evaluation encompasses a range of FMs, categorized as follows: (1) Image Generation Models: This category includes Flux.1-dev (Labs, 2024), Stable Diffusion 3.5 Large (SD3.5) (IT Admin, 2024), and Stable Diffusion XL Base 1.0 (SDXL) (Podell et al., 2023). These models are tasked with generating static images directly from textual or visual inputs. (2) Large Vision-Language Models (LVLMs): We evaluate Deepseek-VL2 (Wu et al., 2024), GLM-4V-9B (GLM et al., 2024), MiniCPM-V2.6 (Yao et al., 2024), Mistral-Small-3.1-24B-Instruct-2503 (Mistral AI, 2025), Phi-3.5-Vision-Instruct (Abdin et al., 2024), Phi-4-Multimodal-Instruct (Abouelenin et al., 2025), Qwen2.5-VL-72B (Team, 2025), GPT-4o (Hurst et al., 2024), Claude 3.7 Sonnet (An-

Table 1: Performance of Diffusion Models, Large Vision Language Models and `v0` on EduVisBench.

| Method | Vis. Type | Maths | | | Physics | | | Chemistry | | | Avg |
|---|---|---|---|---|---|---|---|---|---|---|---|
| | | Easy | Medium | Hard | Easy | Medium | Hard | Easy | Medium | Hard | |
| **Diffusion Model** | | | | | | | | | | | |
| Flux.1-dev | Image | 13.8 | 13.4 | 13.2 | 11.7 | 8.5 | 10.0 | 20.0 | 16.6 | 16.0 | 13.8 |
| SD3.5 | Image | 17.3 | 20.3 | 18.8 | 16.8 | 13.0 | 12.0 | 22.8 | 21.7 | 34.0 | 18.4 |
| SDXL | Image | 17.3 | 23.3 | 25.5 | 18.9 | 15.4 | 24.0 | 33.6 | 30.2 | 24.0 | 21.8 |
| **Large Vision Language Model** | | | | | | | | | | | |
| Deepseek VL2 | Webpage | 20.3 | 17.1 | 15.7 | 17.9 | 17.0 | 20.0 | 16.4 | 13.8 | 14.0 | 17.5 |
| GLM-4V-9B | Webpage | 22.3 | 21.1 | 19.4 | 24.5 | 21.5 | 24.0 | 22.3 | 21.5 | 16.0 | 21.9 |
| MiniCPM-V-2.6 | Webpage | 24.1 | 17.3 | 15.5 | 19.1 | 17.4 | 20.0 | 14.5 | 15.2 | 12.0 | 19.3 |
| Mistral-Small-3.1 | Webpage | 29.1 | 31.6 | 32.2 | 32.3 | 33.5 | 20.0 | 30.6 | 27.5 | 24.0 | 30.2 |
| Phi-3.5 | Webpage | 25.3 | 20.7 | 19.1 | 21.2 | 19.5 | 12.0 | 20.0 | 18.6 | 20.0 | 21.8 |
| Phi-4 | Webpage | 26.1 | 25.1 | 22.9 | 27.8 | 25.5 | 24.0 | 31.2 | 27.5 | 12.0 | 26.4 |
| Qwen2.5-VL-72B | Webpage | 24.3 | 18.1 | 15.8 | 19.7 | 17.1 | 24.0 | 18.2 | 16.4 | 12.0 | 20.0 |
| Claude 3.7 Sonnet | SVG | 61.2 | 26.7 | 23.6 | 18.5 | 16.9 | 14.0 | 47.5 | 47.2 | 18.0 | 42.0 |
| Claude 3.7 Sonnet | Webpage | 56.2 | 57.5 | 55.6 | 44.8 | 42.6 | 24.0 | 61.1 | 60.6 | 64.0 | 54.6 |
| GPT-4o | Webpage | 47.6 | 39.3 | 37.9 | 25.7 | 24.2 | 24.0 | 34.3 | 32.6 | 36.0 | 38.1 |
| GPT-4o | SVG | 36.1 | 19.7 | 19.5 | 13.0 | 12.8 | 4.0 | 30.0 | 27.5 | 22.0 | 26.3 |
| Gemini 2.0 Flash | Webpage | 46.9 | 9.5 | 15.7 | 31.7 | 26.5 | 24.0 | 32.0 | 25.8 | 30.0 | 43.6 |
| **Visualization Agent** | | | | | | | | | | | |
| `v0` | Webpage | 63.0 | 37.6 | 47.2 | 53.3 | 58.5 | 52.0 | 74.7 | 52.8 | 68.0 | 58.2 |

thropic, 2025), and Gemini 2.0 Flash (Mallick & Kilpatrick, 2025). These models are prompted to generate SVG or HTML code, which is then rendered into visual outputs for evaluation. (3) Specialized Visualization Agent: We also assess `v0` (Vercel, 2025), an AI agent specifically designed to create interactive web pages based on instructional content.This diverse selection of models was chosen to represent the current state-of-the-art across different architectural paradigms.

**Evaluation Setups.** During evaluation, all generated visualizations are standardized into image format. For interactive web pages containing buttons, an automated script navigates through all accessible sub-pages, capturing individual screenshots of each. This automated approach ensures that the evaluation is both scalable and free from subjective human bias during the rendering process. Performance is assessed using the evaluation metric described in Section 2.3, where GPT-4o scores the visual outputs based on predefined rubrics, assigning a score from 0 to 5 for each of the five dimensions. The cumulative score (maximum 25 points) is then normalized to a 0-100 scale for standardized reporting and comparison.

**Reliability of GPT-based Scoring.** To validate the reliability of our automatic judge, we compared GPT-based evaluations with human evaluations. Specifically, we selected 50 samples from each subject category (Chemistry, Math, and Physics), and had both GPT and human evaluators independently rate them. Human evaluators were undergraduate

Table 2: Cosine similarity and mean squared error across subjects. Math is the average of Math500 and IllustrativeMath, each with 50 samples.

| Metric | Chemistry | Math | Physics | Average |
|---|---|---|---|---|
| Cosine Similarity ↑ | 0.9742 | 0.9557 | 0.9666 | **0.9655** |
| MSE ↓ | 0.3895 | 0.7093 | 0.6118 | **0.5702** |

students from top universities. We measured agreement using Cosine Similarity and Mean Squared Error (MSE). As shown in Table 2, high agreement—average cosine similarity 0.9655 and MSE 0.5702 across subjects indicates negligible practical discrepancy .

## 4.2 BASELINE BENCHMARKING

The performance of all evaluated baseline models is detailed in Table 1. Across all evaluated models, the average scores indicate significant room for improvement. Diffusion Models generally exhibited the lowest performance, with average scores ranging from 13.8% (Flux.1-dev) to 21.8% (SDXL). This suggests that direct static image generation, while capable of producing visual elements, struggles

substantially with the nuanced requirements of explanatory and guiding visualizations for complex logical problems in our benchmark.

LVLMs typically scored between 17.5% (Deepseek VL2) and 30.2% (Mistral-Small-3.1). Notable exceptions include Gemini 2.0 Flash (43.6%) and Claude 3.7 Sonnet; the latter's significantly better performance with Webpages (54.6%) over SVG (42.0%). GPT-4o also showed a preference for Webpage generation (38.1%) over SVG (26.3%), suggesting that prompting advanced LVLMs for structured interactive webpages can yield more effective visual explanations. Nevertheless, even these top-tier LVLMs face considerable challenges in consistently meeting all of evaluation criteria. The visualization agent v0, specifically designed for webpage generation, achieved the highest average score among all baseline models at 58.2%. This result highlights the advantage of a specialized agent in this task over more general-purpose FMs.

Table 3: Overall comparison of models: left is our EduVisAgent performance, right is the bar chart. EduVisAgent achieves the highest average score among all models.

(a) Performance of our EduVisAgent on EduVisBench.  (b) Comparison of average score across all models.

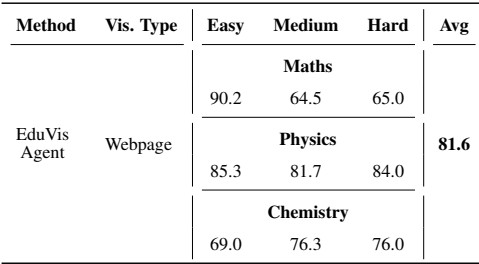
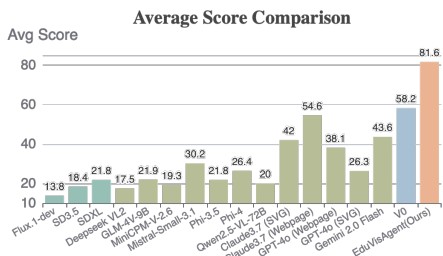

| Method | Vis. Type | Easy | Medium | Hard | Avg |
|--------|-----------|------|--------|------|-----|
| EduVis Agent | Webpage | **Maths** | | | **81.6** |
| | | 90.2 | 64.5 | 65.0 | |
| | | **Physics** | | | |
| | | 85.3 | 81.7 | 84.0 | |
| | | **Chemistry** | | | |
| | | 69.0 | 76.3 | 76.0 | |

### 4.3 PERFORMANCE ANALYSIS OF EDUVISBENCH

Building upon the insights gained from the baseline evaluations, we assessed our proposed multi-agent system, EduVisAgent. The results in Table 3 demonstrate a substantial leap in performance for generating explanatory and logically valuable visualizations for STEM problems. EduVisAgent achieved an impressive overall average score of 81.6%. Specifically, EduVisAgent surpasses the best-performing baseline v0 (58.2%), by 23.4 percentage points (40.2% relative), showing the benefit of the multi-agent architecture and its integration of educational methodologies. Compared to the best performing LVLM (Claude 3.7 Sonnet Webpage at 54.6%) and the top diffusion model (SDXL at 21.8%), the advancement offered by EduVisAgent is even more pronounced. These results clearly indicate that the design principles underlying EduVisAgent, which incorporate a multi-agent structure and pedagogical strategies, effectively address many of the limitations observed in existing generative models.

### 4.4 CASE ANALYSIS

To further illustrate the limitations of existing baselines and how our approach addresses these challenges, we present two case studies in Figure 6. On the left, for a chemistry question, the GPT-4o-generated solution lacks intuitive visualization of the chemical processes, resulting in fragmented information without visual guidance—reflected in a low score of just 28%. In contrast, EduVisAgent begins by displaying background images of the relevant chemical elements, activating students' prior knowledge. This strategy effectively connects abstract chemical concepts to tangible, everyday experiences, a well-established method for enhancing comprehension and retention. It then contextualizes each of the four answer options with real-world scenarios, thereby enhancing students' understanding of the underlying chemical transformations.

Conversely, for the Carnot cycle efficiency physics problem (right side of Figure 6), the Gemini solution presents a single, flawed chart. Its depiction of 300K and 400K temperatures with identical heights introduces visual misinformation, failing to accurately represent data differences and thereby diminishing its pedagogical value. In stark contrast, EduVisAgent employs a multi-agent collaborative approach: it first generates a concrete factory scene to activate students' working memory of the "heat engine" concept. Subsequently, it constructs an accurate Carnot cycle diagram and offers a step-by-step problem breakdown, fostering clear conceptual understanding. Crucially, EduVisAgent

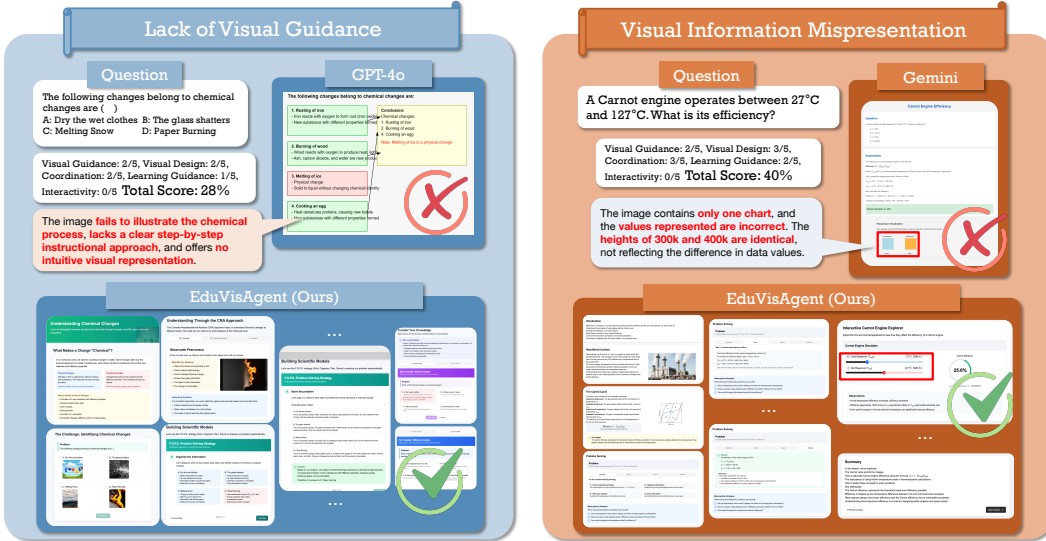

Figure 6: Baseline models versus our EduVisAgent. These examples clearly demonstrate the often poor output quality of baseline models, contrasting sharply with the high-quality, effective visualizations produced by EduVisAgent.

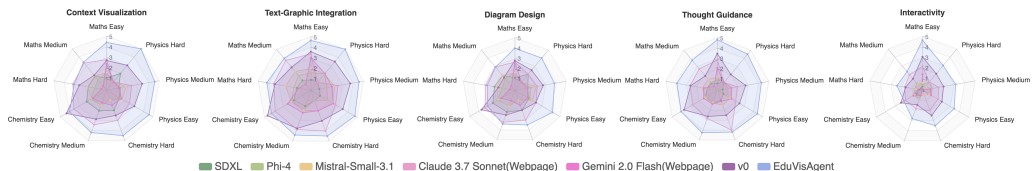

Figure 7: Per-dimension performance comparison across the five evaluation dimensions.

provides interactive visualization components, enabling users to dynamically adjust temperatures via sliders and observe real-time changes in heat engine efficiency. This interactive element transforms the learner from a passive observer into an active participant, which is known to deepen engagement and learning. This interactive engagement significantly facilitates higher-order thinking skills.

Overall, through coordinated multi-agent optimization of image design, instructional structure, and learning pathways, EduVisAgent significantly outperforms traditional single-model approaches in accuracy, guidance, and interactivity.

## 4.5 PER-DIMENSION ANALYSIS

Figure 7 reveals distinct performance profiles for eight high-performing evaluated models. In Context Visualization and Diagram Design, most baselines, including SDXL, Claude 3.7, and v0, exhibit moderate to low scores, often struggling with providing rich situational cues or pedagogically sound visual structures, especially for complex problems. v0 and Claude show relatively better capabilities in Text-Graphic Integration and Thought Guidance compared to other FMs, which generally offer minimal support in these areas. However, all baseline models, including v0, are significantly limited in the Interactivity dimension, primarily due to their output format (static images/SVG or less dynamic webpages). In contrast, our EduVisAgent demonstrates consistently strong performance across all five dimensions. It particularly excels in creating rich context visualizations, well-structured diagram designs, and ensuring seamless text-graphic integration. EduVisAgent also leads by a wide margin in Thought Guidance and Interactivity—the two dimensions where all baselines score lowest—indicating that structured agent coordination is particularly important for these harder-to-achieve pedagogical properties.

## 5 RELATED WORK

**LLM for Pedagogical Assistance.** Foundation models (FMs), including diffusion models and large vision-language models (LVLMs), have been increasingly adopted in educational contexts (Chu et al., 2025; Wang et al., 2024) to support teaching and classroom interactions. EduAgent (Xu et al., 2024) and Teachtune (Jin et al., 2025) enhance the problem-solving process through automated simulations of student-teacher dialogues, collaborative learning, and task-oriented reasoning. Agents such as SEFL (Zhang et al., 2025) and PROF (Nair et al., 2024) synthesize immediate, on-demand feedback to support large-scale instructional scenarios. Domain-specific agents such as MathChat (Wu et al., 2023), NEWTON (Wang et al., 2023b), and MEDCO (Wei et al., 2024) further provide textual explanations tailored to scientific and medical education. While these systems address diverse pedagogical needs, their focus remains largely on text-based interactions (Wu et al., 2023; Xu et al., 2024; Cui et al., 2024), overlooking the critical role of visualization in fostering conceptual understanding and improving learning outcomes (Presmeg, 2006). While valuable, these text-centric systems do not address the large body of educational research highlighting the unique cognitive benefits of visual learning. Despite its pedagogical importance, the capacity of FMs and agents to generate logical, explanatory visual illustrations remains underexplored. EduVisBench is the first benchmark designed to evaluate FMs' ability to produce pedagogically effective, step-by-step visual reasoning, covering 15 visually grounded educational scenarios with multi-level problem sets and multimodal-centric solutions.

**LLM for Scientific Visualization.** While some existing works have preliminarily explored the potential of FMs in supporting visual scaffolding (Podo et al., 2024; Chen et al., 2024c; Pandey & Ottley, 2025; Hong et al., 2025), they are typically fragmented, lack pedagogical grounding, and fail to generalize across diverse educational tasks (Wang et al., 2023a; Ku et al., 2025). For instance, Visual Sketchpad (Hu et al., 2024) attempts to illustrate problem-solving processes with sketches generated from code. However, these visuals are often low in quality, lack logical coherence, and fall short in explanatory depth (Wang et al., 2025). Other approaches like MatplotAgent (Yang et al., 2024), PlotGen (Goswami et al., 2025), and OmniSVG (Yang et al., 2025) use plotting and SVG tools to produce more accurate, data-grounded visualizations. Still, these methods are limited in scope, often addressing only isolated steps rather than providing systematic, end-to-end visual explanations of multi-step problem-solving tasks (Vázquez, 2024; Chen et al., 2024a; 2025b). Our framework, in contrast, is designed to manage the entire pedagogical workflow, from problem deconstruction to the final interactive explanation. To overcome these limitations, we propose a multi-agent collaborative framework, EduVisAgent, that simulates the full learning journey—from initial problem exposure to deep conceptual understanding—by coordinating specialized agents to generate coherent, pedagogically aligned visualizations throughout the reasoning process.

**LLM-based Education Agents.** Recent advancements in LLM-based agents have led to the development of specialized architectures capable of long-horizon planning, tool use, and memory management across a range of real-world domains (Yao et al., 2023; Chan et al., 2024; Chen et al., 2024b; 2025a; Nie et al., 2025; Han et al., 2025; Zhou et al., 2025). In the educational domain, AI agents such as EduAgent (Xu et al., 2024) and Teachtune (Jin et al., 2025) simulate student-teacher dialogues, collaborative learning activities, and task-oriented reasoning to enhance problem-solving instruction. Agents like SEFL (Zhang et al., 2025) and PROF (Nair et al., 2024) generate on-demand feedback for large-scale educational settings, while domain-specific tools such as MathChat (Wu et al., 2023), NEWTON (Wang et al., 2023b), and MEDCO (Wei et al., 2024) provide textual explanations for scientific and medical learning. Despite these advances, limited research has investigated collaborative, multi-agent approaches tailored to educational reasoning and visualization. EduVisAgent is the first multi-agent framework that coordinates specialized agents to support step-by-step pedagogical problem-solving with integrated visual explanations.

## 6 CONCLUSION

This paper addressed the challenge of generating pedagogically meaningful visual explanations with AI systems. We introduced EduVisBench, a benchmark revealing that existing models often produce inadequate visual outputs. This work provides a quantitative baseline for the field, clearly identifying the key areas where current technologies fall short. To overcome this, we proposed EduVisAgent, a collaborative multi-agent framework. Experiments show EduVisAgent significantly outperforms all baselines, demonstrating the potential of agent-based systems for advancing educational visualization.

ETHICS STATEMENT

The primary goal of this research is to advance educational technology by improving the pedagogical quality of AI-generated visualizations, aiming for a positive societal impact. The benchmark developed, EduVisBench, is curated from publicly available and high-quality educational resources, including C-MHChem-Benchmark, high-school-physics, Illustrative Mathematics, and MATH-500. To validate our automated evaluation metric, we conducted a comparative study involving human evaluators, who were undergraduate students from top universities. All data used in the study was handled with care to ensure anonymity and was used solely for the purpose of validating the scoring system. The models and methods proposed are intended for beneficial educational applications. The authors are not aware of any other ethical issues and declare no competing interests or conflicts of interest associated with this research.

REPRODUCIBILITY STATEMENT

To ensure the reproducibility of our work, we have provided detailed descriptions of our methodology and resources. The curation process for our benchmark, EduVisBench, is detailed in Section 2.2, with data sources explicitly cited. Our comprehensive evaluation framework, including the five key dimensions and scoring protocol, is described in Section 2.3 and Section 4.1.The detailed scoring rubrics and the exact prompt used for our GPT-4o-based evaluation are available in Appendix A.2 and Appendix A.3, respectively. The architecture of our proposed EduVisAgent and the roles of each specialized agent are thoroughly explained in Section 3. A complete list of all baseline models and their versions used in our experiments is provided in Section 4.1.

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

# A  APPENDIX

## A.1  VISUALIZATION DISCIPLINES

Table 4 illustrates the disciplines and types in our EduVisBench.

| Discipline | Common Visualization Types |
|---|---|
| *Mathematics* | Number lines, function graphs, and other formalized visual tools. |
| *Physics* | Diagrams involving levers, rigid body motion, forces, and fields. |
| *Chemistry* | Molecular structures and schematic representations of standard laboratory apparatus. |

Table 4: Representative Visualization Types Across Academic Disciplines

## A.2  EVALUATION METRIC

**Visual Scenario Design Guidance**   The category of "Visual Scenario Design Guidance" outlines different levels of visualizing mathematical concepts, progressing from basic text-only representations to highly integrated visual-text formats. Through five defined levels, the framework demonstrates how visual elements can enhance students' understanding and engagement with abstract ideas, guiding instructional designers to gradually enrich scenarios, add annotations, and strengthen contextual connections—ultimately achieving the goal of visually presenting the full flow and conceptual structure of the content.The five levels of Visual Scenario Design Guidance are as follows:

| Level | Description |
|---|---|
| Level 1 | The image contains no scenes or illustrations, presenting only text and formulas. It lacks contextual visual cues, failing to spark interest or connect the concepts to real-life situations. |
| Level 2 | The image includes a single static illustration or low-fidelity mockup with minimal labeling that does not highlight variables or key objects, offering limited context and poor immersion. |
| Level 3 | Multiple static schematic diagrams or sketch-style illustrations appear in the image, labeling core objects, variables, and simple steps, providing basic visual guidance but lacking layered coherence. |
| Level 4 | The image integrates scenario illustrations, storyboard panels, and info-graphics to present the process in multiple views and steps, with annotations and captions guiding students through mapping abstract concepts to context. |
| Level 5 | Storyboard-style illustrations and infographics are fused into a single image, including overview, detailed close-ups, and key pathway diagrams with comprehensive annotations, allowing students to grasp the entire flow and conceptual network at a glance. |

Table 5: Five Levels of Visual Scenario Design Guidance

**Visual Illustration Design**   The category of "Visual Illustration Design" describes progressive levels of visual elements used to support students' systematic understanding of quantities and relationships. It ranges from no visual aids to complex integrated dashboards that deeply connect data and model

structures. Through five levels, the framework guides designers to improve clarity, coherence, and contextual richness of visual illustrations, enhancing students' analytic and comparative abilities.

| Level | Description |
|-------|-------------|
| Level 1 | The image contains no charts, axes, or flow diagrams—only text. Without embedded visual tools, students cannot systematically organize or analyze quantities and relationships. |
| Level 2 | The image presents a static number line and colored bar chart with complete scales and a legend, helping students gain a basic understanding of numerical changes. However, it lacks comparison and contextual layering. |
| Level 3 | The image presents a static number line and colored bar chart with complete scales and legends, helping students grasp basic numerical changes visually, though comparison and context layering are absent. |
| Level 4 | The image combines number lines, flowcharts, infographics, and arrow annotations; multiple visuals are juxtaposed or overlaid to show processes and variable changes for a coherent modeling view. |
| Level 5 | The image presents a dashboard-style visualization integrating axes, bar charts, flow diagrams, heatmaps, etc., with linked elements that deeply visualize data relationships and model structure. |

Table 6: Five Levels of Visual Illustration Design

**Text–Illustration Coordination**  The category of "Text–Illustration Coordination" describes levels of alignment and integration between textual content and visual elements within images. This progression ranges from complete disconnection to seamless fusion, enabling students to effectively map and synthesize text, formulas, and graphics. The framework guides designers in strengthening links between verbal and visual information to enhance comprehension and structural understanding.

| Level | Description |
|-------|-------------|
| Level 1 | Text and illustrations in the image are completely disconnected, with no labels, legends, or connectors—students cannot use visuals to understand text or formulas. |
| Level 2 | Text occasionally prompts "see diagram" or "refer to the illustration," but the image lacks legends or clear labels, so mapping between text and graphics remains ambiguous. |
| Level 3 | Text descriptions and image elements share consistent numbering, color blocks, or arrows linked to a simple legend, explaining core symbols and variables to support initial mapping. |
| Level 4 | Text paragraphs are laid out alongside corresponding visuals within the same image, with detailed legends and color-coded annotations enabling simultaneous reading and mapping. |
| Level 5 | Text, formulas, and legends are fully integrated in one image, using consistent colors, numbering, and layered layout to achieve seamless text–graphic fusion for complete structural understanding. |

Table 7: Five Levels of Text–Illustration Coordination

**Learning Thought Guidance**  The category of "Learning Thought Guidance" describes the progressive inclusion of visualized problem-solving strategies and reflective cues in images. From presenting

only problem statements to complex integrated dashboards, this framework guides designers to scaffold students' strategic thinking and metacognitive reflection through visual tools, enabling deeper reasoning and transfer of learning.

| Level | Description |
|---|---|
| Level 1 | The image offers no visualized problem-solving guidance, showing only the problem statement and formulas, leaving students without strategic cues or reflection prompts. |
| Level 2 | The image embeds a simple flowchart or two title-style hints (e.g., "Identify problem type," "Check result"), but the flowchart is overly simplistic and hints lack hierarchical detail. |
| Level 3 | The image displays a step-by-step flowchart template with key thinking nodes and self-check checkpoints, leaving annotation space for students to visually record their reasoning. |
| Level 4 | The image combines a near-transfer exercise with a comparative thought diagram, visually highlighting strategy differences so students can apply existing reasoning to a new context. |
| Level 5 | The image fuses near- and far-transfer exercises, concept mind maps, and a reflection panel into a dashboard-style layout, allowing students to review and extend their problem-solving network visually. |

Table 8: Five Levels of Learning Thought Guidance

**Interactivity and Personalized Support**    The category of "Interactivity and Personalized Support" outlines levels of incorporating feedback, hints, and tailored assistance into images, evolving from static presentations to dynamic, student-responsive visual supports. This framework encourages designers to embed interactive elements that adapt to learner needs, promoting engagement and personalized problem-solving.

| Level | Description |
|---|---|
| Level 1 | The image includes no feedback or support components—only a static problem statement and answer field—offering no hints, examples, or error cues and resulting in a nonresponsive visual. |
| Level 2 | The image shows fixed hint boxes (e.g., "Hint: draw a number line," "Hint: check rounding"), but hints are not tailored to student responses, limiting personalized guidance. |
| Level 3 | The image integrates multiple static correction tips and example solution modules (common mistakes and standard approaches), which students can reference visually but without intelligent recommendations. |
| Level 4 | The image presents example solution workflows, text hints, and a common-errors analysis section highlighted with color blocks and arrows, providing diverse visual support in a single layout. |
| Level 5 | The image displays a comprehensive visual support panel with difficulty suggestions, personalized hints, worked examples, and extension resource links, enabling students to select tailored guidance directly from the visual layout. |

Table 9: Five Levels of Interactivity and Personalized Support

## A.3 EVALUATION PROMPT

The instructional web page evaluation prompt is structured as follows:

---

**Evaluation Prompt**

As a **professional evaluator of instructional web pages**, your task is to determine whether the generated web page meets expectations across five specific categories.

**Instructions:**
- Assign an **integer score from 0 to 5** for each of the five categories (1–5).

- **0** = completely missing or extremely poor
  **5** = fully meets the highest standard

- Evaluation should be based solely on the specified aspect: *{category}*.
  The definition of *{category}* is: *{description}*.

- **Do not include any explanation, justification, or additional commentary. Refusing to provide a score is not allowed.**

---

**Evaluation Output Format**

```
{{RATING: {"1":score, "2":score, "3":score, "4":score, "5":score}}}
```

---

## A.4 GENERATION PROMPTS FOR BASELINE MODELS

The following prompts are used to instruct baseline models to generate instructional visualizations for multiple-choice questions.

---

**HTML Webpage Generation Prompt**

Please generate a fully structured and styled HTML webpage for the following {subject} multiple-choice question, with a focus on clearly presenting the problem, visualizing key scientific concepts, and explaining the reasoning behind the correct choice.
The page should include:
- A prominently displayed question section.
- A clearly formatted list of answer choices (A, B, C, D).
- A step-by-step explanation section that helps users understand why the correct answer is right, and why the others are not. This section may include:
  – Diagrams or illustrations (e.g., molecules, environmental impact visuals),
  – Charts or data comparisons (e.g., particulate levels),
  – Flowcharts or labeled process diagrams,
  – Any other visual representation that supports comprehension.
- A clear highlight of the final correct answer (e.g., a visual cue or box).
Requirements:
- Use modern CSS styling (inline `<style>` block or external).
- Ensure layout is responsive and readable across devices.
- Use semantic HTML structure with headings and subheadings (`<h1>`, `<h2>`, etc.).
- Render scientific symbols or formulas correctly (e.g., MathJax or KaTeX).
- Visualization libraries such as Chart.js, D3.js, or SVG may be used to enhance explanations.
Question: {question}
Please output the full HTML + CSS + JavaScript code only, without any extra explanation or comments.

---

---

**SVG Diagram Generation Prompt**

Generate a **stand-alone SVG diagram** that visually explains the following question as a reasoning task.
The SVG must include:

- The full question text;

- Key reasoning steps, visual annotations, or illustrations that aid in understanding or solving the problem;

- If applicable, highlight the final answer or conclusion clearly.

Requirements:

- Output only a single `<svg>...</svg>` block, with no extra text outside it;

- SVG width between 800–1200px, layout should be clean and adaptive;

- Use `<text>` elements with readable font sizes for all text;

- Use arrows, symbols, and diagrams if they help communicate the solution process;

- You may omit multiple-choice options if not relevant.

Question: {question}

---

**Visualization Agent Generation Prompt**

Your task is to generate a complete webpage solution for the following problem. The page should include:

- Introduce a scenario to engage students.

- Explain the problem background and provide a step-by-step walkthrough of the solution.

- Give another similar problem to assess students' understanding.

Question: {question}

## A.5 MULTI-AGENT SYSTEM PROMPTS

### A.5.1 TASK PLANNING AGENT PROMPT

---

**System Prompt**

Transform learner's question into structured instructional task.
1. **Scenario & Understanding**: Real-world context + givens/goals/constraints
2. **Solution Strategy**: Analysis + step-by-step solution + reasoning
3. **Transfer Tasks**:
   - Near-transfer: same structure, minor changes
   - Far-transfer: different surface, same logic
4. **FOPS Structure** (per step):
   - **F**: Problem type    **O**: Diagram/equation structure    **P**: Solution path    **S**: Execute + verify
   - Specify: goal, action, concepts, pitfalls
5. **UI Layout**: Shadcn/UI structure + sections + meta-prompt placement + math rendering

**JSON Output Format**

```
{ "scenario": { "context": "...", "givens": ["g1","g2"],
    "goals": ["goal1"], "constraints": ["c1"] },
  "solution_strategy": { "analysis": "...",
    "steps": [{ "step_number": 1, "fops_label": "F/O/P/S",
      "description": "What to do", "reasoning": "Why" }] },
  "transfer_tasks": {
    "near_transfer": { "problem": "...", "explanation": "..." },
    "far_transfer":  { "problem": "...", "explanation": "..." } },
  "instructional_steps": [{ "step_number": 1, "goal": "...",
    "action": "...", "concepts": ["c1","c2"], "pitfalls": ["m1"] }],
  "ui_layout_suggestion": { "structure": "Page flow",
    "key_components": ["Card","Accordion","Alert","Table"],
    "content_organization": "...", "math_rendering": "KaTeX/MathJax" }
}
```

---

**User Prompt**

Give your analysis to the following question in JSON format.    {Question}

---

### A.5.2 CONCEPTUAL MAPPING AGENT PROMPT

---

**System Prompt**

Map concepts using **CRA framework** (Concrete–Representational–Abstract).
1. **Concrete**: Objects, quantities, situations (directly experienced)
2. **Representational + Visual Design**:
   - Tools: number lines, bar graphs, diagrams, organizers, sketches, tables
   - Visual design: form, purpose, concrete→abstract bridge; avoid decorative
3. **Abstract**: Formulas, principles, theorems + connection to concrete
4. **Think Aloud**: Verbalization prompts

**JSON Output Format**

```
{ "cra_mapping": [{ "step_number": 1, "step_name": "...",
    "concrete": { "entities": ["obj1","qty1"], "description": "..." },
    "representational": [{ "tool": "Number line/etc",
      "purpose": "Why it helps", "represents": "What it shows",
      "visual_design": { "form": "...", "elements": ["..."],
        "bridge": "How it bridges concrete to abstract" } }],
    "abstract": { "constructs": ["formula1","principle1"],
      "connection": "How abstract connects to concrete" },
    "think_aloud": ["prompt1", "prompt2"] }]
}
```

---

**User Prompt**

According to the following planning, give your analysis in JSON format.     {Task Planning JSON}

### A.5.3  REASONING DECOMPOSITION AGENT PROMPT

**System Prompt**

Decompose reasoning using design **scaffolded practice**.
1. **Step Guidance**: Action (what learner does) + link to CRA concepts + **visual support** (which CRA visual + interaction)
2. **Practice Activities**: Types (fill-in, choice, judgment, explanation) + gradual release strategy
3. **Math**: LaTeX in $...$ or $$...$$

**JSON Output Format**

```
{ "fops_reasoning": [{ "action": "What learner does",
    "concepts": ["c1","c2"], "scaffolding_needed": true,
    "scaffolding_notes": "Why and how to scaffold",
    "visual_support": { "needed": true,
      "which_visual": "CRA mapping ref", "interaction": "..." } }],
  "practice_activities": [{ "activity_number": 1,
    "name": "Activity name", "type": "fill-in/choice/judgment/explanation",
    "task": "Problem or task content",
    "gradual_release": "How independence builds" }]
}
```

**User Prompt**

According to the following planning and conceptual mapping, give your analysis in JSON format.
{Task Planning JSON}     {Conceptual Mapping JSON}

### A.5.4  METACOGNITIVE REVIEWER PROMPT

**System Prompt**

Generate metacognitive prompts for **monitor, evaluate, regulate**.
1. **Reflective Prompts**: Check understanding, comprehension, strategy
2. **Self-Questioning**: "What did I do?" / "Why does this work?" / "What did I miss?" / "How does this connect?" / "What if...?"
3. **Self-Correction**: Checkpoints + error identification
4. **Organize by Phase**: **Before** (understanding + planning) → **During** (monitoring + checking) → **After** (evaluating + reflecting)
5. **Math**: LaTeX in $...$ or $$...$$

**JSON Output Format**

```
{ "metacognitive_prompts": {
    "before_solving": ["Understanding check 1", "Planning prompt 1"],
    "during_solving": ["Step monitoring 1", "Strategy adjustment 1"],
    "after_solving": ["Verification 1", "Reflection 1", "Improvement 1"]
  },
  "general_strategies": [{ "strategy": "Self-monitoring technique",
    "when_to_use": "During which phase",
    "how_to_apply": "Specific actions" }]
}
```

> **User Prompt**
>
> According to the following planning, conceptual mapping and reasoning decomposition, give your analysis in JSON format.
> {Task Planning JSON}
> {Conceptual Mapping JSON}
> {Reasoning Decomposition JSON}

### A.5.5 VISUALIZATION AGENT PROMPT

> **System Prompt**
>
> As the **Visual Representation Specialist**, generate the visualization teaching webpage of the given question.
>   1. **Abstract Visualization Tools**: Encourage number lines, bar graphs, object diagrams, organizers, sketches, and data tables.
>   2. **Avoid Decorative Imagery**: No photorealistic or cartoon-style images; visuals must remain schematic and pedagogical.
>   3. **Conceptual Mapping**: Explicitly explain how each visual corresponds to the underlying abstract concept (e.g., number line for quantity change).
>   4. **UI Integration**: Specify how visuals should appear in the web UI (placement, sequencing, interaction).
>   5. **Rendering**: Math rendered with KaTeX/MathJax.

> **User Prompt**
>
> Using all of the given data, generate a visualized teaching webpage for the following question.
> {Question}    {Task Planning JSON}    {Conceptual Mapping JSON}
> {Reasoning Decomposition JSON}    {Metacognitive Reviewer JSON}

## B  ADDITIONAL RESULTS

### B.1  ABLATION STUDIES ON EDUVISAGENT

We conduct ablation studies on the multi-agent teaching system in Table 10.

We compare the overall Total score of the complete system (**Full**) against six variants: **-TP** (without Task Planning), **-CM** (without Conceptual Mapping), **-RD** (without Reasoning Decomposition), **-MR** (without Metacognitive Review), **-VIS** (Using Claude 3.7 Sonnet as the underlying model while keeping the same multi-agent pipeline), and **Single** (a single-agent baseline that uses the same set of prompts without explicit modularization or inter-agent coordination).

**Impact of Removing Individual Modules.**    The ablation results reveal clear and systematic evidence that each module in the multi-agent architecture contributes a distinct cognitive function whose removal leads to measurable degradation. Across all subjects and difficulty levels, dropping any single module results in a consistent decline of approximately 4–6 points relative to the full model, indicating that no component is redundant.

**Task Planning (-TP).** Removing the Task Planning module reduces the system's ability to structure the solution trajectory at the outset. This manifests in weaker performance particularly on Medium and Hard problems, where multi-step planning is essential. Without this stage, the problem-solving process becomes more linear and less globally coherent, leading to incremental reasoning errors that accumulate throughout the solution.

**Conceptual Mapping (-CM).** The Conceptual Mapping module provides the system with an intermediate representational scaffold—a way to organize domain concepts, formulas, and symbolic relations before detailed reasoning begins. Its removal produces one of the largest degradations across domains, showing that the system heavily relies on this symbolic "schema construction" phase.

Table 10: Ablation study on EduVisAgent. **Full** denotes our complete multi-agent system. **Single** denotes a single-agent baseline using all prompts without modularization. The highest scores are in **bold**.

| Subject | Difficulty | Full | -TP | -CM | -RD | -MR | -VIS | Single |
|---------|-----------|------|------|------|------|------|------|--------|
| **Chemistry** | Easy | **69.00** | 64.60 | 63.80 | 63.00 | 64.20 | 64.80 | 59.30 |
| | Medium | **76.27** | 71.87 | 71.07 | 70.27 | 71.47 | 72.07 | 67.42 |
| | Hard | **76.00** | 71.60 | 70.80 | 70.00 | 71.20 | 71.80 | 66.85 |
| **Physics** | Easy | **85.33** | 80.94 | 80.14 | 79.34 | 80.54 | 81.14 | 76.10 |
| | Medium | **81.71** | 77.31 | 76.51 | 75.71 | 76.91 | 77.51 | 72.05 |
| | Hard | **84.00** | 79.60 | 78.80 | 78.00 | 79.20 | 79.80 | 74.32 |
| **Maths** | Easy | **90.20** | 85.80 | 85.00 | 84.20 | 85.40 | 86.00 | 81.05 |
| | Medium | **64.50** | 60.10 | 59.30 | 58.50 | 59.70 | 60.30 | 55.27 |
| | Hard | **65.00** | 60.60 | 59.80 | 59.00 | 60.20 | 60.80 | 53.88 |

Without CM, the reasoning tends to jump directly into procedural steps without establishing the underlying conceptual structure, which leads to misapplied rules, inconsistent variable usage, and missing constraints.

**Reasoning Decomposition (-RD).** This module creates explicit fine-grained segmentation of logical steps, and its absence consistently yields the largest drop among all ablations. Without RD, the model tends to condense multi-step reasoning into single large leaps, increasing the likelihood of hidden errors that are never surfaced or corrected. RD therefore serves as the backbone of reliable multi-hop reasoning, ensuring that intermediate steps are interpretable, verifiable, and less prone to compounding mistakes.

**Metacognitive Review (-MR).** The MR module acts as an internal critic, performing self-evaluation and error checking. Removing MR reduces the model's ability to detect calculation inconsistencies, missing assumptions, or logical conflicts within its own output. While the primary reasoning path remains intact, the absence of this reflective layer leads to small but systematic accuracy losses, especially on tasks requiring unit checking, boundary conditions, or verification of derived results.

**Model Replacement (-VIS).** Replacing the underlying model with `Claude 3.7 Sonnet` while preserving the multi-agent structure produces a moderate drop across categories. The decline is smaller than removing core cognitive modules, demonstrating that the multi-agent pipeline itself contributes substantial performance stability. However, it also indicates that the pipeline and the underlying model are jointly responsible for peak performance.

**Single-Agent Baseline.** The **Single** baseline, which collapses all prompts into a single monolithic agent, performs the worst across all conditions. Its scores are consistently below every multi-agent ablation, averaging 8–15 points lower than the full system. This sharp decline highlights that the benefits of the multi-agent design arise not merely from the prompts themselves, but from the explicit architectural separation of planning, conceptual structuring, stepwise reasoning, and self-review.

### B.2 EVALUATION ON NON-STEM DATASETS

Table 11: Performance of Visualization Agents on EduVisBench.

| Method | Vis. Type | Prehistory | Sociology | Avg |
|--------|-----------|-----------|-----------|-----|
| v0 | Webpage | 61.8 | 68.0 | 64.9 |
| EduVisAgent | Webpage | 79.4 | 84.8 | 82.1 |

To further explore EduVisAgent's performance in non-STEM disciplines, we selected 100 problems from MMLU's prehistory and sociology categories to evaluate EduVisAgent's adaptability to narrative and conceptual reasoning tasks. Results in Table 11 show that EduVisAgent significantly outperforms

the current SOTA baseline (v0) by 17.2 points, demonstrating that our multi-agent framework effectively generalizes to humanities education.

### B.3 EVALUATION ROBUSTNESS

We tested two other evaluators, Gemini 2.5 pro and Claude 3.7 Sonnet to score visualization results of SDXL, GPT-4o(SVG) and EduVisAgent. The results in Table 12 show consistent scoring patterns across different evaluator models. While there are minor numerical variations in the absolute scores, the overall trends and relative rankings remain highly consistent with the GPT-4o results.

Table 12: Evaluated by GPT-4o, Gemini 2.5 Pro and Claude 3.7 Sonnet on EduVisBench.

| Method | Vis. Type | GPT-4o | Gemini 2.5 Pro | Claude 3.7 Sonnet |
|---|---|---|---|---|
| SDXL | Image | 21.8 | 18.9 | 24.7 |
| GPT-4o | SVG | 25.4 | 26.3 | 27.4 |
| EduVisAgent | Webpage | 78.9 | 81.6 | 84.4 |

## C ADDITIONAL MATHEMATICS EXAMPLES IN EDUVISBENCH

Figure 8a, 8b, 8c and 8d presents four representative problems from the Hard subset (MATH-500) in EduVisBench, illustrating the range and difficulty of the mathematical reasoning challenges in the benchmark.

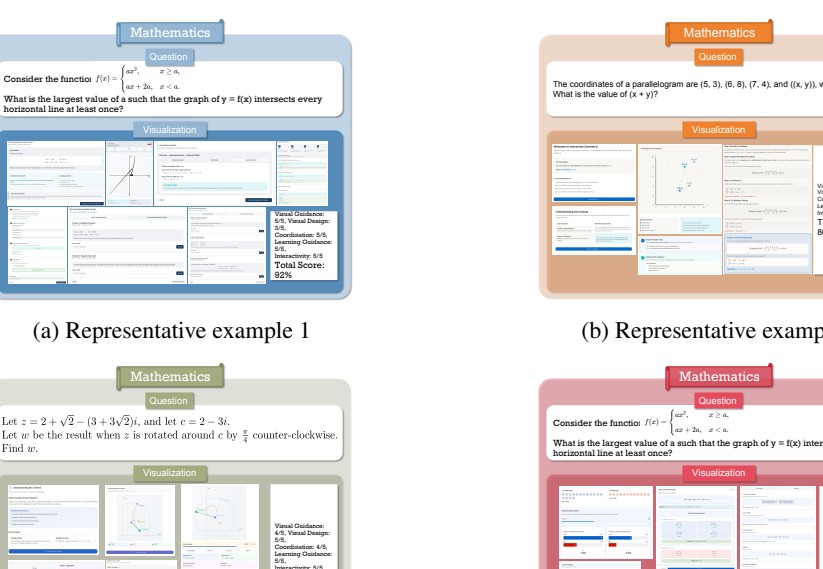

(a) Representative example 1      (b) Representative example 2

(c) Representative example 3      (d) Representative example 4

Figure 8: Representative examples from the Hard subset (MATH-500) of EduVisBench.

## D REAL-WORLD DEPLOYMENT OPTIMIZATION

For real-world deployment, we identify several optimization directions: (1) reducing token consumption through more efficient prompt engineering, (2) optimizing the metacognitive reasoning depth to balance quality and efficiency, and (3) implementing caching mechanisms for frequently used educational concepts. Additionally, the system could operate asynchronously. Teachers prepare materials in advance rather than real-time generation, making latency less critical.

# E FULL PROMPT AND INTERMEDIATE STRUCTURES FOR THE ANGULAR VELOCITY EXAMPLE

We present the complete prompt and intermediate output structures for all five agents in the angular velocity example.

**Agent 1: Task Planning**

```
task_planning:
  scenario:
    context: A wheel is rotating with a given angular acceleration and initial angular
      velocity. The task is to determine its angular velocity after a specific time.
    givens:
      - Angular acceleration (\alpha) = 4 rad/s^2
      - Initial angular velocity (\omega_0) = 2 rad/s
      - Time (t) = 5 seconds
    goals: [Find the angular velocity (\omega) after 5 seconds]
    constraints: [Use the kinematic equation for angular motion]
  solution_strategy:
    analysis: This problem involves angular motion with constant angular acceleration.
      The angular velocity can be calculated using \omega = \omega_0 + \alpha t.
    steps:
      - step_number: 1
        fops_label: F: Constant acceleration / O: \omega = \omega_0 + \alpha t / P: Substitute / S: Solve
        description: Use \omega = \omega_0 + \alpha t. Substitute the given values.
        reasoning: This equation relates all given quantities. Directly applicable.
      - step_number: 2
        fops_label: F: Substitution / O: \omega = \omega_0 + \alpha t / P: Arithmetic / S: Verify
        description: Substitute \omega_0=2, \alpha=4, t=5 into \omega = 2 + (4 x 5).
        reasoning: Substituting the values allows direct calculation.
      - step_number: 3
        fops_label: F: Final calculation / O: \omega = 2+20 / P: Simplify / S: Verify
        description: Simplify: \omega = 2 + 20 = 22 rad/s.
        reasoning: Ensures the final value is correct and matches given options.
```

```
  transfer_tasks:
    near_transfer:
      problem: A wheel rotates with angular acceleration 3 rad/s^2. If its initial
        angular velocity is 5 rad/s, what is its angular velocity after 4 seconds?
      explanation: Same structure as original but with different numerical values.
    far_transfer:
      problem: A car accelerates uniformly from rest at 3 m/s^2. What is its velocity after 6 s?
      explanation: Involves linear motion instead of angular, but same kinematic logic.
  instructional_steps:
    - step_number: 1
      goal: Understand the problem and identify given values.
      action: Extract \alpha, \omega_0, and t from the problem statement.
      concepts: [Angular motion, Kinematic equations]
      pitfalls: [Confusing angular with linear velocity, Misidentifying given values]
    - step_number: 2
      goal: Apply the kinematic equation for angular velocity.
      action: Write \omega = \omega_0 + \alpha t and substitute the given values.
      concepts: [Kinematic equation for angular motion]
      pitfalls: [Incorrect substitution, Forgetting initial angular velocity]
    - step_number: 3
      goal: Calculate the final angular velocity.
      action: Perform the arithmetic calculation to find \omega.
      concepts: [Basic arithmetic, Angular velocity]
      pitfalls: [Arithmetic errors, Misinterpreting the result]
  ui_layout_suggestion:
    structure: Step-by-step instructional layout with sections for scenario, solution, practice.
    key_components: [Card, Accordion, Alert, Table]
    content_organization: Scenario at top, solution strategy with step-by-step, practice at end.
    math_rendering: Use KaTeX or MathJax for equations like \omega = \omega_0 + \alpha t.
```

**Agent 2: Conceptual Mapping**

```
conceptual_mapping:
  cra_mapping:
    - step_number: 1
      step_name: Understand the problem and identify the given values
      concrete:
        entities: [wheel, angular acceleration, initial angular velocity, time]
        description: A rotating wheel with given angular acceleration and initial
          angular velocity over a specific time.
      representational:
        - tool: Table
          purpose: Organize the given values clearly for reference.
          represents: \alpha, \omega_0, and t.
          visual_design:
            form: Simple table with columns for quantity, symbol, and value.
            elements: [Headers for quantity/symbol/value, Rows for \alpha/\omega_0/t]
            bridge: Transition from problem statement to identifying needed variables.
      abstract:
        constructs: [Kinematic equation: \omega = \omega_0 + \alpha t]
        connection: Relates given quantities (\alpha, \omega_0, t) to unknown \omega.
      think_aloud: [What quantities are given?, What is the goal?]
```

```
    - step_number: 2
      step_name: Apply the kinematic equation for angular velocity
      concrete:
        entities: [angular velocity, kinematic equation]
        description: Using \omega = \omega_0 + \alpha t to calculate angular velocity.
      representational:
        - tool: Equation
          purpose: Show relationship between \omega, \omega_0, \alpha, and t.
          represents: The kinematic equation for angular motion.
          visual_design:
            form: Standard mathematical notation.
            elements: [\omega = \omega_0 + \alpha t]
            bridge: Links abstract formula to concrete problem via substitution.
        - tool: Dynamic animation (optional)
          purpose: Illustrate how \omega changes over time with constant acceleration.
          represents: Growth of angular velocity as time progresses.
          visual_design:
            form: Graph showing \omega increasing linearly with time.
            elements: [Time on x-axis, Angular velocity on y-axis]
            bridge: Visualizes the relationship described by the equation.
      abstract:
        constructs: [Substitution of values into \omega = \omega_0 + \alpha t]
        connection: Substituting concrete values into abstract formula.
      think_aloud: [What equation to use?, How to substitute values?]
```

```
    - step_number: 3
      step_name: Calculate the final angular velocity
      concrete:
        entities: [arithmetic calculation, final angular velocity]
        description: Performing the calculation to find the final angular velocity.
      representational:
        - tool: Step-by-step arithmetic
          purpose: Break down calculation into smaller steps for clarity.
          represents: The process of solving \omega = 2 + (4 x 5).
          visual_design:
            form: Sequential steps showing each part of the calculation.
            elements: [Substitute values, Multiply \alpha and t, Add to \omega_0]
            bridge: Connects abstract substitution to concrete result.
        - tool: Alert or feedback box
          purpose: Highlight common errors in arithmetic or substitution.
          represents: Potential pitfalls in the calculation process.
          visual_design:
            form: Colored box with tips or warnings.
            elements: [Error examples, Corrective suggestions]
            bridge: Prevents errors and reinforces correct methods.
      abstract:
        constructs: [Final result: \omega = 22 rad/s]
        connection: The calculated result is the concrete realization of the formula.
      think_aloud: [What is the next step?, Does the result make sense?]
```

**Agent 3: Reasoning Decomposition**

```
reasoning_decomposition:
  fops_reasoning:
    - action: Extract \alpha, \omega_0, and t from the problem statement.
      concepts: [Angular motion, Kinematic equations]
      scaffolding_needed: true
      scaffolding_notes: Learners may confuse angular with linear velocity or
        misidentify values. Provide a table to organize.
      visual_support:
        needed: true
        which_visual: Table for organizing given values
        interaction: Draggable labels to match quantities with symbols and values
    - action: Write \omega = \omega_0 + \alpha t and substitute the given values.
      concepts: [Kinematic equation for angular motion]
      scaffolding_needed: true
      scaffolding_notes: Learners may struggle with identifying the correct equation
        or substituting correctly. Provide guided substitution exercise.
      visual_support:
        needed: true
        which_visual: Equation with placeholders for substitution
        interaction: Reveals for each substitution step
    - action: Perform the arithmetic calculation to find \omega.
      concepts: [Basic arithmetic, Angular velocity]
      scaffolding_needed: true
      scaffolding_notes: Arithmetic errors are common. Break down calculation into
        smaller steps and provide feedback on errors.
      visual_support:
        needed: true
        which_visual: Step-by-step arithmetic breakdown
        interaction: Dynamic reveals for each calculation step
```

```
practice_activities:
  - activity_number: 1
    name: Identify Given Values | type: fill-in
    task: Fill in the table with values for \alpha, \omega_0, and t.
    gradual_release: Guided examples -> independent identification.
  - activity_number: 2
    name: Substitute Values into Equation | type: choice
    task: Choose correct substitution for \omega = \omega_0 + \alpha t given
      \alpha=4, \omega_0=2, t=5.
    gradual_release: Hints and feedback -> independent substitution.
  - activity_number: 3
    name: Calculate Angular Velocity | type: judgment
    task: Perform \omega = 2 + (4 x 5) and verify correctness.
    gradual_release: Step-by-step guidance -> independent calculation with feedback.
  - activity_number: 4
    name: Near Transfer Problem | type: explanation
    task: A wheel with \alpha=3 rad/s^2, \omega_0=5 rad/s. Find \omega after 4s.
    gradual_release: Guided steps -> independent problem-solving.
  - activity_number: 5
    name: Far Transfer Problem | type: explanation
    task: A car accelerates from rest at 3 m/s^2. Velocity after 6 seconds?
    gradual_release: Comparison to angular motion -> independent solving.
```

**Agent 4: Metacognitive Reviewer**

```
metacognitive_reviewer:
  metacognitive_prompts:
    before_solving:
      - What quantities are given in the problem, and what do they represent?
      - What is the goal of the problem, and how can I achieve it?
    during_solving:
      - Am I using the correct equation for this problem?
      - Have I substituted the values correctly into the equation?
    after_solving:
      - Does my result make sense given the context of the problem?
      - What could I improve in my approach to solving similar problems?

  general_strategies:
    - strategy: Self-monitoring technique
      when_to_use: During the substitution and calculation phases
      how_to_apply:
        Pause after each step to verify the accuracy of substitutions and calculations.
    - strategy: Error-checking routine
      when_to_use: After completing the calculation
      how_to_apply:
        Revisit the problem statement and verify that the result aligns with the given values and context.
    - strategy: Visualization
      when_to_use: During the understanding phase
      how_to_apply:
        Organize the given values in a table or diagram to clarify relationships.
```

**Agent 5: Visualization Page Design**

```
visualization_page:
  layout_overview:
    structure: Step-by-step instructional layout with clearly separated sections.
    sections:
      - {id: scenario_overview, title: Scenario Overview, content: Problem statement + given values}
      - {id: step_by_step_solution, title: Step-by-Step Solution, content: Interactive visuals}
      - {id: practice_problems, title: Practice Problems, content: Near/far-transfer tasks}
      - {id: reflection_prompts, title: Reflection Prompts, content: Metacognitive questions}
    ui_components: [Cards, Accordion, Alerts, Interactive visuals (tables/equations/animations)]
  static_visuals:
    step_1: {goal: Organize given values, type: table, columns: [Quantity, Symbol, Value],
      rows: [\alpha, \omega_0, t], interaction: Drag-and-drop labels}
    step_2: {goal: Make equation explicit, equation: \omega = \omega_0 + \alpha t,
      substitution: \omega = 2+(4x5), interaction: Click placeholders to reveal values}
    step_3: {goal: Transparent arithmetic, breakdown: [4x5=20, 2+20=22], form: flowchart}
  dynamic_behavior:
    angular_velocity_graph:
      purpose: Show \omega increasing linearly with constant \alpha.
      axes: {x: Time 0-5s, y: \omega 0-22 rad/s}
      behavior: {start: 2, end: 22 rad/s}, interaction: Animate on load.
    stepwise_substitution:
      purpose: Visualize substitution in \omega = \omega_0 + \alpha t.
      placeholders: [\omega_0=2, \alpha=4, t=5], timing: fade-in per reveal.
  user_interactions:
    interactive_table: Drag-and-drop to match quantities, symbols, values.
    equation_substitution: Click placeholders to reveal substituted values.
    arithmetic_breakdown: "Next step" revealing 4x5=20, then 2+20=22.
```

# F    QUALITATIVE SUCCESS AND FAILURE CASES

## SUCCESS CASES

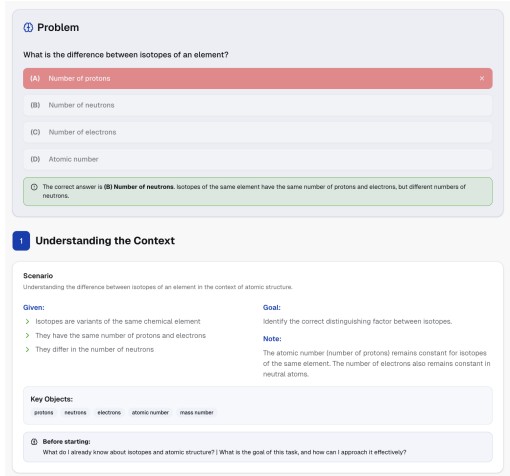

Figure 9: Success case 1 illustrating coherent multi-agent collaboration.

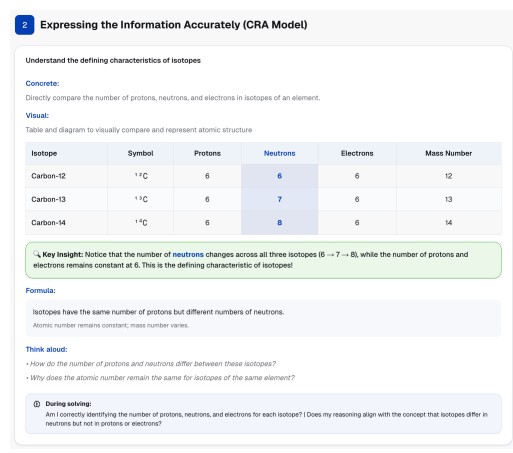

Figure 10: Success case 2 illustrating coherent multi-agent collaboration.

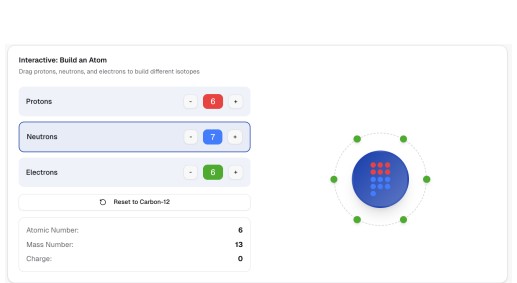

Figure 11: Success case 3 illustrating coherent multi-agent collaboration.

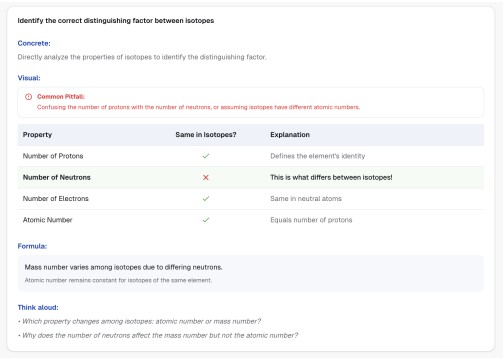

Figure 12: Success case 4 illustrating coherent multi-agent collaboration.

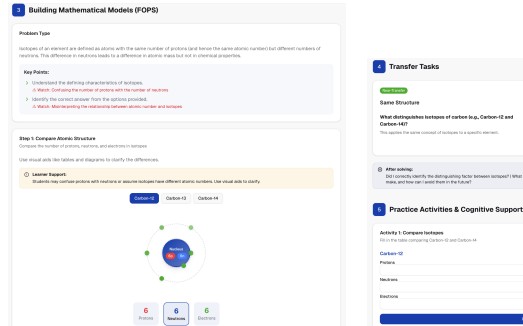

Figure 13: Success case 5.

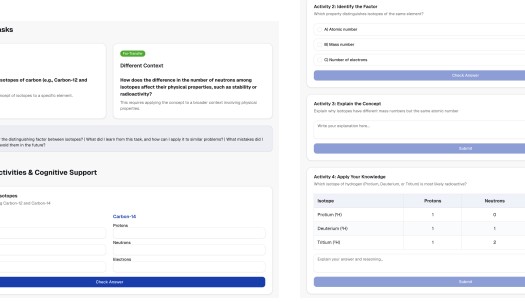

Figure 14: Success case 6.

Figure 15: Success case 7.

FAILURE CASE: ERROR ACCUMULATION IN THE MULTI-AGENT PIPELINE

We illustrate a representative failure mode in which an early planning error propagates through the multi-agent pipeline. In **Step 1**, the `task_planning` agent proposes an inappropriate `ui_layout_suggestion`—elements such as `Card`, `Accordion`, `Alert`, and `Table` bear no meaningful relation to the underlying concept of angular velocity. In **Step 2**, downstream reasoning agents correctly infer the need for a rotating-wheel visualization, yet the erroneous layout suggestion continues to impose irrelevant UI constraints. In **Step 3**, the Visualization Agent attempts to comply with upstream directives, producing non-physics-related components instead of correct angular-motion visuals. By **Step 4**, the multi-agent chain has amplified the initial mistake: despite correct domain inference, the final UI is structurally misaligned. The **final outcome** (Step 5) shows that accumulated constraint errors prevent the system from producing any meaningful visualization related to angular velocity, demonstrating error propagation in multi-agent settings.

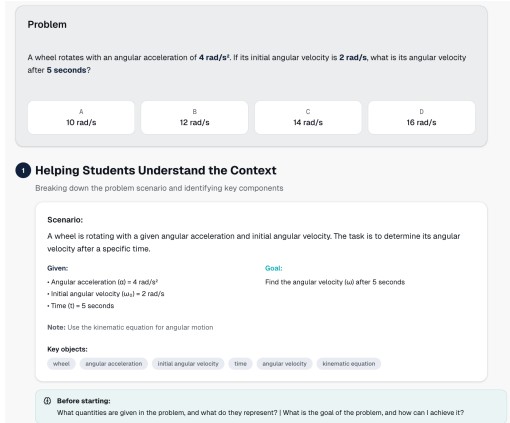

Figure 16: Failure case step 1: inappropriate layout suggestion.

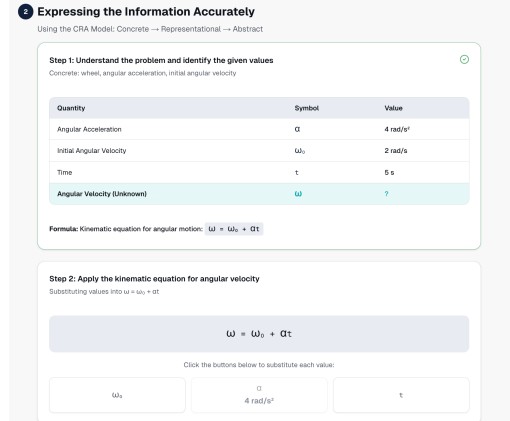

Figure 17: Failure case step 2: erroneous constraints propagate.

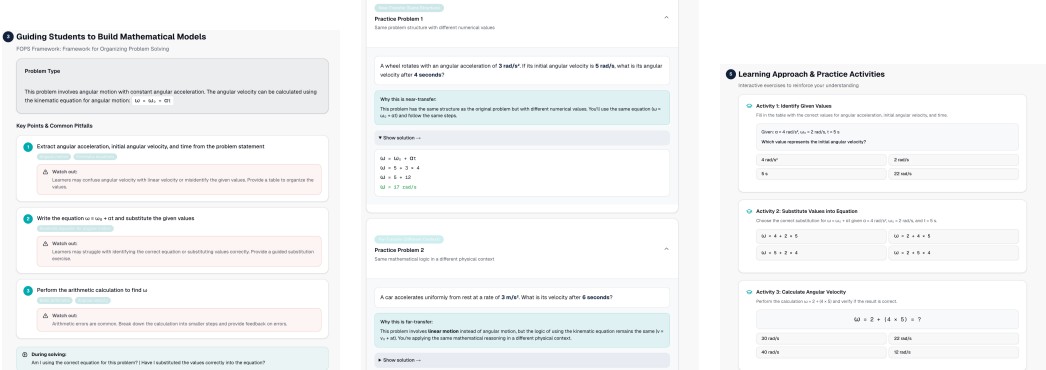

Figure 18: Failure case step 3: non-physics components.

Figure 19: Failure case step 4: structurally misaligned UI.

Figure 20: Failure case step 5: no meaningful visualization.

