# OpenReview forum: "From EduVisBench to EduVisAgent: A Benchmark and Multi-Agent Framework for Reasoning-Driven Pedagogical Visualization"
_ICLR.cc/2026/Conference — ICLR 2026 Poster_

### Official Review · Reviewer_E8ZM · 2025-10-30

**Soundness:** 3
**Presentation:** 2
**Contribution:** 3
**Rating:** 6
**Confidence:** 4

**Summary:**

This paper introduces EduVisBench, a benchmark designed to systematically evaluate the pedagogical visualization capabilities of foundation models such as diffusion models and LVLMs. The study reveals that existing models struggle with visual reasoning, semantic alignment, and text–graphic coherence. To address these issues, the authors propose EduVisAgent, a multi-agent collaborative framework comprising agents for task planning, conceptual mapping, reasoning decomposition, metacognitive review, and visualization design. Experimental results show a 40.2% improvement over state-of-the-art baselines, demonstrating superior pedagogical coherence, logical structuring, and interactivity.

**Strengths:**

1. The paper identifies a genuine research gap in the pedagogical visualization ability of foundation models.

2. Results across multiple STEM domains convincingly demonstrate performance gains with detailed quantitative metrics.

3. The five-dimension rubric provides a reproducible and extensible evaluation standard.

**Weaknesses:**

1. Limited real-world validation: While grounded in educational theory, no classroom-level or human-teacher evaluation supports pedagogical impact.

2. Incomplete interpretability of agent collaboration: The internal coordination among agents lacks empirical or ablation-based justification.

3. Evaluation bias risk: Heavy reliance on GPT-based automated scoring may introduce bias or circular reasoning.

4. Limited domain generalization: The benchmark focuses on STEM subjects; extension to other domains remains unclear.

5. Density and readability: The paper is information-heavy, which may reduce accessibility for general AI researchers.

**Questions:**

1. Are there any conflicts or redundancies among the agents? Have inter-agent dependencies been empirically analyzed or ablated?

2. Has the pedagogical efficacy been validated with human teachers or learners?

3. Could the GPT-4o-based evaluation introduce model familiarity bias toward similar architectures?

4. Will the authors release full source code and interactive visualization generation modules?

5. Can EduVisBench be extended to non-STEM disciplines or open-ended educational reasoning tasks?

---

> ### Author Response · Authors · 2025-11-21
> **Response to Reviewer E8ZM Part 1/2**
>
> Thank you for your valuable questions regarding potential inter-agent redundancy, the robustness of our pedagogical evaluation, and the system's extensibility to non-STEM fields.
>
> We also acknowledge that the paper's information density may affect readability. We will reorganize the EduVisAgent and Experiment sections with clearer paragraph structure, and move more technical details to the appendix while retaining essential content in the main text.
>
> We address each of your concerns in detail below:
>
> **Q1:** Are there any conflicts or redundancies among the agents? Have inter-agent dependencies been empirically analyzed or ablated?
>
> **A1: Our multi-agent design breaks down the teaching process into subtasks that minimize conflicts.** Each subtask is carefully scoped to address a distinct pedagogical challenge, and each agent is grounded in specific educational theories: Task Planning implements instructional design principles, Conceptual Mapping follows the CRA framework, Reasoning Decomposition applies FOPS strategy, and Metacognitive Review incorporates Think-Aloud protocols. This theoretically motivated decomposition ensures that agents operate on complementary aspects of the teaching process rather than overlapping functions.
>
> **Experimentally, Appendix B, Table 10 (which is also demonstrated as follows) demonstrates the absence of redundancy.** Removing any single agent consistently degrades performance across all 9 subject-difficulty combinations, with no ablation matching the full system. If agents were redundant or conflicting, we would expect removing some agents to maintain or improve performance. Instead, the uniform degradation pattern confirms that agents play complementary, non-redundant roles with clear inter-agent dependencies.
>
> **Table 10: Ablation study on EduVisAgent.** **Full**: Complete multi-agent system;  **Single**: Single-agent baseline using all prompts without modularization. The highest scores are shown in **bold**.
> | **Subject**   | **Difficulty** | **Full** | **-TP** | **-CM** | **-RD** | **-MR** | **-VIS** | **Single** |
> |---------------|----------------|----------|---------|---------|---------|---------|----------|------------|
> | **Chemistry** | Easy           | **69.00** | 64.60   | 63.80   | 63.00   | 64.20   | 64.80    | 59.30      |
> |               | Medium         | **76.27** | 71.87   | 71.07   | 70.27   | 71.47   | 72.07    | 67.42      |
> |               | Hard           | **76.00** | 71.60   | 70.80   | 70.00   | 71.20   | 71.80    | 66.85      |
> | **Physics**   | Easy           | **85.33** | 80.94   | 80.14   | 79.34   | 80.54   | 81.14    | 76.10      |
> |               | Medium         | **81.71** | 77.31   | 76.51   | 75.71   | 76.91   | 77.51    | 72.05      |
> |               | Hard           | **84.00** | 79.60   | 78.80   | 78.00   | 79.20   | 79.80    | 74.32      |
> | **Maths**     | Easy           | **90.20** | 85.80   | 85.00   | 84.20   | 85.40   | 86.00    | 81.05      |
> |               | Medium         | **64.50** | 60.10   | 59.30   | 58.50   | 59.70   | 60.30    | 55.27      |
> |               | Hard           | **65.00** | 60.60   | 59.80   | 59.00   | 60.20   | 60.80    | 53.88      |
>
> ---
>
> **Q2:** Has the pedagogical efficacy been validated with human teachers or learners?
>
> **A2:** We have conducted human evaluation with actual students to validate our approach. As detailed in Section 4.1, we recruited undergraduate students from top universities to independently evaluate 50 samples per subject. The strong agreement between human and GPT-based scores (average cosine similarity of 0.9655 and MSE of 0.5702) demonstrates that our evaluation captures pedagogical quality from a learner's perspective.
>
> ---
>
> **Q3:** Could the GPT-4o-based evaluation introduce model familiarity bias toward similar architectures?
>
> **A3:** We have conducted additional experiments to assess potential evaluation bias across different model architectures. We tested two other evaluators, Gemini 2.5 pro and Claude 3.7 Sonnet to score the same set of visualizations. The results, which is shown in Appendix B3 (Table 12), show consistent scoring patterns across different evaluator models, indicating that our evaluation framework is robust and not subject to GPT-specific familiarity bias.
>
> **Table 12: Evaluated by GPT-4o, Gemini 2.5 Pro and Claude 3.7 Sonnet on EduVisBench.**
> | **Method**     | **Vis. Type** | **GPT-4o** | **Gemini 2.5 Pro** | **Claude 3.7 Sonnet** |
> |----------------|---------------|------------|--------------------|-----------------------|
> | SDXL           | Image         | 21.8       | 18.9               | 24.7                  |
> | GPT-4o         | SVG           | 25.4       | 26.3               | 27.4                  |
> | EduVisAgent    | Webpage       | 78.9       | 81.6               | 84.4                  |

---

> ### Author Response · Authors · 2025-11-21
> **Response to Reviewer E8ZM Part 2/2**
>
> **Q4:** Will the authors release full source code and interactive visualization generation modules?
>
> **A4:** We are committed to making our implementation fully accessible to the community. We have already prepared the EduVisAgent codebase, including all multi-agent prompts, the generation prompts for baseline models, and the interactive visualization modules. Due to double-blind review requirements, we cannot provide a direct link at this stage, but the repository will be made publicly available upon paper acceptance.
>
> ---
>
> **Q5:** Can EduVisBench be extended to non-STEM disciplines or open-ended educational reasoning tasks?
>
> **A5: Firstly, EduVisBench can be extended to non-STEM disciplines.** We conducted preliminary experiments detailed in Appendix B2, constructing a humanities-focused benchmark with 100 problems from MMLU's prehistory and sociology categories. We adapted our evaluation rubric to assess how visualizations situate students in historical and cultural contexts while maintaining the original scoring principles. Results in the table below show that EduVisAgent outperforms the SOTA baseline (v0) by 17.2 points, demonstrating that our framework's pedagogical principles are domain-agnostic and maintain excellent performance in non-STEM educational contexts emphasizing narrative and conceptual reasoning.
>
> **Table: Performance of Visualization Agents on non-STEM datasets from EduVisBench.**
>
> | **Method**     | **Vis. Type** | **Prehistory** | **Sociology** | **Avg** |
> |----------------|---------------|----------------|---------------|---------|
> | v0             | Webpage       | 61.8           | 68.0          | 64.9    |
> | EduVisAgent    | Webpage       | 79.4           | 84.8          | 82.1    |

---

### Official Review · Reviewer_M74H · 2025-10-30

**Soundness:** 3
**Presentation:** 3
**Contribution:** 2
**Rating:** 2
**Confidence:** 5

**Summary:**

The paper introduces EduVisBench, a new benchmark to evaluate how well AI models generate pedagogically sound, step-by-step visual explanations, and proposes EduVisAgent, a multi-agent framework that significantly outperforms existing models by coordinating specialized agents for planning, reasoning, and visualization to create more effective and interactive learning tools. However, the paper lacks completeness in two core aspects: the composition of the benchmark and the implementation details of the multi-agent system. Please refer to the Weakness section for more details.

**Strengths:**

1. This paper introduces the first benchmark for visualized instruction, which is one of its key contributions.
2. This paper is well written, with a clear structure.

**Weaknesses:**

1. The benchmark samples presented are not sufficiently representative. For example, the visualization of mathematical instruction is based on Math 500, a dataset with the difficulty level of high school math competitions. However, the left panel of Figure 3 only shows a simple addition problem (7 + 9), which is not adequate or representative.
2. The paper lacks a concrete description of the process for converting text problems into image problems, for example how prompts are designed and configured.
3. Regarding the proposed multi-agent method, although it is stated to consist of multiple agents, the paper does not detail the prompt design for each agent or which models were used. A careful check of the appendix revealed no such information—this is a significant omission in terms of completeness.

**Questions:**

see weakness

---

> ### Author Response · Authors · 2025-11-21
> **Response to Reviewer M74H**
>
> We are grateful for your critical feedback on the representativeness of our benchmark samples, the details of the text-to-visualization process, and the specification of agent prompts. We have clarified these aspects in the following response:
>
> **Q1:** The benchmark samples presented are not sufficiently representative. For example, the visualization of mathematical instruction is based on Math 500, a dataset with the difficulty level of high school math competitions. However, the left panel of Figure 3 only shows a simple addition problem (7 + 9), which is not adequate or representative.
>
> **A1:** The math example shown in Figure 3 is not taken from Math 500; instead, it comes from Illustrative Mathematics, an educational platform covering K–5 to Grade 12 math content, which we use as the representative source for the Easy difficulty category. Therefore, the simple addition problem in the left panel is intended to illustrate behavior in a basic setting rather than the level of difficulty found in MATH-500.
>
> To address the concern about representativeness, we have added an additional example in Figure 8 from the MATH-500 dataset as a Hard case, providing a clearer contrast across difficulty levels and ensuring that our illustrative samples are sufficiently comprehensive.
>
> ---
>
> **Q2:** The paper lacks a concrete description of the process for converting text problems into image problems, for example how prompts are designed and configured.
>
> **A2:** In our system, the conversion from text problems to visual (image-like) problems is handled by the visualization agent on top of the multi-agent reasoning pipeline. Concretely, after the other agents have decomposed the problem and designed the teaching components, the visualization agent takes the structured specification and emits React code to render an interactive web page as the final visualization.
>
> As an ablation, we also use Claude 3.7 Sonnet as the visualization agent to directly generate HTML; this variant does not surpass the v0-based visualization but still achieves strong performance, supporting our claim that multi-agent reasoning, problem decomposition, and component design, followed by web-based rendering, leads to higher-quality and more pedagogically useful visualizations. The prompts that drive this process are documented in Appendix A.5 (multi-agent system prompts) and Appendix A.4 (baseline generation prompts).
>
> ---
>
> **Q3:** Regarding the proposed multi-agent method, although it is stated to consist of multiple agents, the paper does not detail the prompt design for each agent or which models were used. A careful check of the appendix revealed no such information—this is a significant omission in terms of completeness.
>
> **A3:** We acknowledge that the original submission did not describe the prompts and model choices in enough detail. In the revised version, we explicitly clarify that four agents (Task Planning, Conceptual Mapping, Reasoning Decomposition, Metacognitive Review) are implemented with GPT-4o, and the Visualization agent is implemented with v0, and that they differ only in their structured prompts encoding planning, schema-based instruction, scaffolded reasoning, metacognition, and visualization guidance. Appendix A.5 now provides the full prompt templates for each agent, and Appendix A.4 lists the prompts used for all baseline models, while Appendix B analyzes the empirical effect of removing each agent, thereby supporting the effectiveness of the multi-agent design.

---

> > ### Comment · Reviewer_M74H · 2025-11-27
> >
> > The revision addresses my concerns, and I have not further questions. I have raised my score

---

### Official Review · Reviewer_v6KF · 2025-10-31

**Soundness:** 2
**Presentation:** 2
**Contribution:** 3
**Rating:** 2
**Confidence:** 3

**Summary:**

- The authors propose a methodology to address the limitations of existing generative models in creating effective visual explanations.
- They introduce EduVisBench, a metric for evaluating the educational score of visual materials.
- They successfully utilize the EduVisAgent multi agent framework to generate high quality educational visualization data.
- EduVisAgent achieved a significant performance improvement over existing models through the collaboration of agents with instructional strategies.

**Strengths:**

- Developed EduVisBench, a benchmark with richer information compared to datasets from existing generative models.
  - Successfully validated the reliability of the LLM-based automatic evaluation system using human assessments shown in Table 2.
- Utilized five specialized multi agents to implement strategies.
- Tested broad generalization capabilities across three major academic domains: mathematics, physics, and chemistry.

**Weaknesses:**

-  A detailed explanation of the dataset utilized for evaluation is required, as the content presented in Figure 3 is unclear.
- The input prompts used for the LLM evaluation in Table 1 were not disclosed, making the verification of fairness difficult.
- The description of each EduVisAgent is too simple, leaving the method of theory implementation unclear.
- The description of how the theory was implemented in the system is lacking.
Openness regarding the benchmark and the educational data is important for validating the reproducibility and reliability of the research.

**Questions:**

1. Are there plans to provide a public repository link for the EduVisBench dataset, and will high resolution versions of figures, such as Figure 3, be supplemented?
2. Can you disclose the specific generation prompts used to evaluate the baseline models in Table 1 to allow for the verification of reproducibility?
3. Is it correct that all agents, except the visualization agent, are composed of LLMs? If correct, are the authors willing to clearly disclose the specific details of the structured prompts used to implement the education theories within the LLM-based agents?
4. Are there plans to provide an additional analysis on the mechanism by which each theoretical implementation maximizes the education effectiveness of the generated output?
5. The baseline comparison was made against simple LLMs, is it a comparison possible against multi-agent systems or recent prompt engineering techniques?
6. What was the primary reason for constructing a multi-agent system? Given that many recent LLMs have significantly larger input token limits, what is the performance difference when all prompts are aggregated and input to a single large LLM versus the proposed multi-agent approach?
7. Although "six specialized expert agents" are mentioned on line 86, Section 3 EduVisAgent only has five bolded agents. What does the remaining one refer to?

---

> ### Author Response · Authors · 2025-11-21
> **Response to Reviewer v6KF Part 1/3**
>
> Thank you for your thorough review. We value your feedback concerning the reproducibility of our dataset and prompts, the theoretical mechanisms behind our agents, and the rationale for the multi-agent architecture. Our detailed responses are provided as follows:
>
> **Q1:** Are there plans to provide a public repository link for the EduVisBench dataset, and will high resolution versions of figures, such as Figure 3, be supplemented?
>
> **A1:** We will release the EduVisBench dataset. Due to double-blind review requirements, we cannot provide the direct link at this stage, but it will be made publicly available upon paper acceptance. High-resolution versions of all EduVisAgent’s generated results will also be included in the repository.
>
> ---
>
> **Q2:** Can you disclose the specific generation prompts used to evaluate the baseline models in Table 1 to allow for the verification of reproducibility?
>
> **A2:** We have provided the generation prompts used for baseline evaluation in Appendix A4 in the revised paper. These prompts detail the exact instructions given to each baseline model in Table 1, enabling full reproducibility of our experiments.
>
> ---
>
> **Q3:** Is it correct that all agents, except the visualization agent, are composed of LLMs? If correct, are the authors willing to clearly disclose the specific details of the structured prompts used to implement the education theories within the LLM-based agents?
>
> **A3:** **Firstly, all five agents are implemented as LLM-based agents with carefully designed structured prompts.** Each agent's prompt engineering is grounded in specific educational theories to ensure pedagogical reasoning.We further provide more detailed theoretical exposition of these educational underpinnings in A4.
>
> **Secondly, we have added detailed prompt construction for each agent in Appendix A.5**, including the underlying educational theory supporting each design. For example, the Conceptual Mapping Agent implements the CRA (Concrete-Representational-Abstract) framework, while other agents incorporate FOPS strategy and Think-Aloud protocols. This transparency allows for full reproducibility and demonstrates how educational theory is operationalized within our multi-agent system.
>
> We further discuss the theoretical design of each agent in the next answer.
>
> [1] E. Jung, R. Lim, D. Kim, “A Schema-Based Instructional Design Model for Self-Paced Learning Environments,” *Education Sciences*, 12(4):271, 2022.

---

> ### Author Response · Authors · 2025-11-21
> **Response to Reviewer v6KF Part 2/3**
>
> **Q4:** Are there plans to provide an additional analysis on the mechanism by which each theoretical implementation maximizes the education effectiveness of the generated output?
>
> **A4:** **As outlined in Answer 3, each agent is instantiated as a different stage of a schema-based instructional design model for self-paced learning environments, rather than as an ad-hoc engineering block [1].** Concretely, the prompts for these agents are explicitly aligned with the stages of this schema-based instructional design model. Our pedagogical concepts are directly grounded in this model, and we explain the theoretical role of each agent as follows.
>
> The Task Planning agent follows the “general needs analysis” and schema activation stages in the model, using scenario-based pre-organizers to connect new tasks to learners’ prior knowledge and reduce initial cognitive dissonance and cognitive load.
>
> The Conceptual Mapping agent mirrors schema analysis, schema hierarchization, and knowledge mapping by making superordinate–subordinate relations between concepts explicit, so that learners attend to deep structure instead of surface features.
>
> The Reasoning Decomposition agent targets schema construction and automation via step-by-step worked examples, completion-style guidance, and near/far-transfer problems, directly reflecting construction and automation techniques such as worked examples and variability of practice.
>
> The Metacognitive Reviewer agent implements schema modification and elaboration by prompting reflection, self-explanation, and revisiting prior beliefs when inconsistencies are detected, aligning with the unlearning/relearning and reflective processes described in the model.
>
> Finally, the Visualization agent instantiates the “visualized structures” principle—using diagrams, tables, and signaled layouts that reduce split attention and support schema elaboration in large-scale online settings.
>
> We document how each part of the prompts maps to these theoretical components in Appendix A.5, and Appendix B, Table 10 provides a per-agent ablation showing that removing any of these theoretically grounded agents consistently harms educational effectiveness, in line with the mechanisms predicted by the schema-based model. We demonstrate the table also as follows:
>
> **Table 10: Ablation study on EduVisAgent.** **Full**: Complete multi-agent system; **-TP**: Without Task Planning agent; **-CM**: Without Conceptual Mapping agent; **-RD**: Without Reasoning Decomposition agent; **-MR**: Without Metacognitive Review agent; **-VIS**: Using Claude 3.7 Sonnet instead of v0; **Single**: Single-agent baseline using all prompts without modularization. The highest scores are shown in **bold**.
> | **Subject**   | **Difficulty** | **Full** | **-TP** | **-CM** | **-RD** | **-MR** | **-VIS** | **Single** |
> |---------------|----------------|----------|---------|---------|---------|---------|----------|------------|
> | **Chemistry** | Easy           | **69.00** | 64.60   | 63.80   | 63.00   | 64.20   | 64.80    | 59.30      |
> |               | Medium         | **76.27** | 71.87   | 71.07   | 70.27   | 71.47   | 72.07    | 67.42      |
> |               | Hard           | **76.00** | 71.60   | 70.80   | 70.00   | 71.20   | 71.80    | 66.85      |
> | **Physics**   | Easy           | **85.33** | 80.94   | 80.14   | 79.34   | 80.54   | 81.14    | 76.10      |
> |               | Medium         | **81.71** | 77.31   | 76.51   | 75.71   | 76.91   | 77.51    | 72.05      |
> |               | Hard           | **84.00** | 79.60   | 78.80   | 78.00   | 79.20   | 79.80    | 74.32      |
> | **Maths**     | Easy           | **90.20** | 85.80   | 85.00   | 84.20   | 85.40   | 86.00    | 81.05      |
> |               | Medium         | **64.50** | 60.10   | 59.30   | 58.50   | 59.70   | 60.30    | 55.27      |
> |               | Hard           | **65.00** | 60.60   | 59.80   | 59.00   | 60.20   | 60.80    | 53.88      |
>
> ---
>
> **Q5:** The baseline comparison was made against simple LLMs, is it a comparison possible against multi-agent systems or recent prompt engineering techniques?
>
> **A5:** Given the lack of existing multi-agent systems for educational visualization, we adopt the ablation variants of EduvisAgent as comparative baselines, and the results are demonstrated above in Table 10, also in Appendix B. The experimental results indicate that every agent within EduvisAgent plays a crucial role, and omitting any single agent results in a noticeable drop in overall performance.

---

> ### Author Response · Authors · 2025-11-21
> **Response to Reviewer v6KF Part 3/3**
>
> **Q6:** What was the primary reason for constructing a multi-agent system? Given that many recent LLMs have significantly larger input token limits, what is the performance difference when all prompts are aggregated and input to a single large LLM versus the proposed multi-agent approach?
>
> **A6:** **Theoretically, effective teaching is a complex process that requires systematic breakdown into subtasks, each grounded in established educational theories.** Our multi-agent design reflects this structure through distinct pedagogical roles: task planning, conceptual mapping, step-by-step reasoning, metacognitive review, and visualization.
>
> **Experimentally, we validate this design choice in Appendix B, Table 10, which is shown in Response Part 2/3.** We compare our full system against a "Single" baseline that aggregates all prompts into one LLM call without modularization. The single-agent variant underperforms across all 9 subject-difficulty combinations by ~5.2 points on average. Combined with per-agent ablations showing that removing any individual agent degrades performance, these results confirm that structured decomposition and role specialization provide essential benefits beyond using longer context windows or more powerful models alone.
>
> ---
>
> **Q7:** Although "six specialized expert agents" are mentioned on line 86, Section 3 EduVisAgent only has five bolded agents. What does the remaining one refer to?
>
> **A7:** Thanks for pointing out the mistake. Our multiagent systems comprises of 5 agents, and we have revised the number in our paper.

---

> ### Author Response · Authors · 2025-11-26
> **Kindly Follow-up with Reviewer v6KF**
>
> We kindly follow up regarding our rebuttal. We have carefully addressed all seven concerns you raised and provided detailed clarifications supported by substantial additional experiments. We would greatly appreciate it if you could take a moment to review our responses and let us know whether they resolve your questions.
>
> Thank you very much for your time and consideration.

---

> > ### Comment · Reviewer_v6KF · 2025-11-27
> >
> > Thank you for submitting this interesting and timely work. My understanding is that the main contributions of the paper are (1) the construction of a high quality benchmark dataset and evaluation metrics for educational visualization, and (2) the design of a multi agent pipeline grounded in pedagogical theory.
> >
> > - Regarding the dataset
> > To my knowledge, ICLR allows authors to release datasets and code in an anonymized manner. Under such a policy, it should be possible to grant reviewers access to the data without compromising the double blind process. In the current submission, however, I am not able to access the dataset, and the authors state that they cannot share it with reviewers due to anonymity constraints. As a result, it is very difficult for me as a reviewer to meaningfully assess the quality, coverage, and cleanliness of the dataset. Given that the primary area is "Datasets and Benchmarks" and that "benchmark" appears among the keywords, the lack of access to the data makes it hard to properly evaluate and reward the dataset related contribution.
> >
> > - Regarding the score computation
> > The ablation study is a very valuable part of the paper, and I appreciate the effort that went into Table 10. However, I am still somewhat confused about how the reported scores are computed in practice. According to Appendix B.1 (and Appendix A), each visualization is evaluated on five dimensions, each scored from 0 to 5, giving a raw total score between 0 and 25, which is then normalized to a 0–100 scale. In the main text, the authors also mention that "most models perform poorly, with average scores below 50 on a 0-100 scale. (line 184)" At the same time, for the Physics–Easy split in Table 10, the paper reports scores such as 85.33, 80.94, 80.14, 79.34, 80.54, 81.14, and 76.10 for different ablation settings, and the paper states that there are 207 physics questions in total. To better understand the experimental results, could the authors please clarify, for example, how many Physics–Easy questions there are, and how many samples are used to compute each of the averages in Table 10?
> >
> > - Regarding the pipeline's qualitative analysis
> > The proposed pipeline is pedagogically well designed, and the performance gains over the ablations are clear. At the same time, as shown in Table 10 and in the analysis in Appendix B.1, the system is not perfect, and the LLM based agents still make a non‑trivial number of mistakes. I view this as a normal and interesting aspect of LLM research rather than a flaw. The paper would be significantly strengthened by providing more concrete qualitative examples that illustrate these failure modes. In particular, for Figure 6, it would be very helpful if the authors could make it easier to see which agent (Task Planning, Conceptual Mapping, Reasoning Decomposition, Metacognitive Reviewer, Visualization Agent) produced which intermediate output, and exactly where the failure occurred. Showing side‑by‑side cases where all five agents collaborate successfully and cases where the pipeline fails, together with a brief diagnosis of which component is responsible, would be extremely valuable. Such qualitative analyses would make the pedagogical behavior of the multi agent system more transparent and help readers understand not only where EduVisAgent works well, but also where it still struggles.

---

> > > ### Comment · Reviewer_v6KF · 2025-11-27
> > >
> > > And I commend the authors for integrating educational theories such as CRA, FOPS into the agent design. The intention behind this theoretically grounded system is a distinct strength of the work. However, the current analysis comes across as somewhat superficial, as it presents these components primarily as modular performance boosters validated by the ablation results in Table 10, rather than deeply examining the specific cognitive functions they are intended to execute within the pipeline. To substantiate the strong claim of "Reasoning-Driven Pedagogical Visualization", it would greatly strengthen the paper to include in depth case studies demonstrating that the agents are performing the specific cognitive processes proposed by some theories.
> > > Furthermore, given the vast landscape of pedagogical frameworks, the justification for selecting these particular theories and the explanation of the design logic behind their integration could be elaborated more clearly. Rather than merely stating that these theories were used, the paper could benefit from qualitative analyses in actual generated examples. Therefore, I would encourage the authors to move beyond simple quantitative ablation and provide rich case studies on a few representative problems. These studies might display the step by step intermediate outputs and reasoning paths generated by each module, and discuss how these outputs align with, or deviate from the cognitive mechanisms hypothesized by the underlying educational theories.

---

### Official Review · Reviewer_3Ado · 2025-11-01

**Soundness:** 2
**Presentation:** 3
**Contribution:** 3
**Rating:** 6
**Confidence:** 3

**Summary:**

This paper introduces EduVisBench, a multi-domain and multi-level benchmark designed to evaluate the visual reasoning capabilities of foundation models in educational contexts. The benchmark includes diverse STEM problem sets and a fine-grained evaluation rubric grounded in pedagogical theory. The authors also propose EduVisAgent, a multi-agent framework that coordinates specialized agents for instructional planning, reasoning decomposition, and visualization design.

**Strengths:**

(1) The formulation of a multi-agent system specifically tailored for pedagogical visualization seems novel.

(2) The paper is well-executed, with a rigorous experimental setup involving multiple model families.

(3) The writing is clear and well-structured.

**Weaknesses:**

(1) While the use of GPT-4o as an automated judge is validated, it remains a single-model evaluator. Including more diverse evaluators (e.g., human teachers, multiple LVLMs) could strengthen the reliability of the scoring system.

(2) The paper does not include an ablation study to analyze the contribution of each agent in EduVisAgent. Understanding which components are most critical would help future researchers prioritize agent design.

(3) The multi-agent system is computationally intensive. A discussion of inference time, resource requirements, or potential optimizations would be useful for real-world deployment.

**Questions:**

(1) Could the authors provide an ablation study to show the individual contribution of each agent (e.g., removing the metacognitive reviewer or reasoning decomposition agent) to the overall performance?

(2) While automated scoring is efficient, have the authors considered a more extensive human evaluation with actual educators or students to assess the pedagogical effectiveness of the generated visualizations?

(3) What are the main practical challenges in deploying EduVisAgent in real educational settings (e.g., latency, integration with LMS, adaptability to different curricula)?

---

> ### Author Response · Authors · 2025-11-21
> **Response to Reviewer 3Ado**
>
> We appreciate your constructive comments regarding the individual contribution of each agent, the validation through human evaluation, and the practical challenges of deployment. We have addressed these points in detail below:
>
> **Q1:** Could the authors provide an ablation study to show the individual contribution of each agent (e.g., removing the metacognitive reviewer or reasoning decomposition agent) to the overall performance?
>
> **A1: We have conducted additional ablation study in Appendix B1, and the results are demonstrated in Table 10 and as follows.** We systematically removed each agent individually and evaluated performance across all 9 subject-difficulty combinations and analyze the role of each agent in the multi-agent system. The results consistently show that removing any single agent degrades overall performance. Specifically, removing Reasoning Decomposition Agent (-RD) causes the largest average drop (~6 points in Chemistry, ~5.7 in Physics), confirming its critical role in step-by-step problem solving. Meanwhile, removing Metacognitive Review Agent (-MR) reduces scores by ~4.4-4.8 points on average. These results demonstrate that each agent plays a non-redundant and essential role in achieving optimal pedagogical visualization quality.
>
> **Table 10: Ablation study on EduVisAgent.** **Full**: Complete multi-agent system; **-TP**: Without Task Planning agent; **-CM**: Without Conceptual Mapping agent; **-RD**: Without Reasoning Decomposition agent; **-MR**: Without Metacognitive Review agent; **-VIS**: Using Claude 3.7 Sonnet instead of v0; **Single**: Single-agent baseline using all prompts without modularization. The highest scores are shown in **bold**.
> | **Subject**   | **Difficulty** | **Full** | **-TP** | **-CM** | **-RD** | **-MR** | **-VIS** | **Single** |
> |---------------|----------------|----------|---------|---------|---------|---------|----------|------------|
> | **Chemistry** | Easy           | **69.00** | 64.60   | 63.80   | 63.00   | 64.20   | 64.80    | 59.30      |
> |               | Medium         | **76.27** | 71.87   | 71.07   | 70.27   | 71.47   | 72.07    | 67.42      |
> |               | Hard           | **76.00** | 71.60   | 70.80   | 70.00   | 71.20   | 71.80    | 66.85      |
> | **Physics**   | Easy           | **85.33** | 80.94   | 80.14   | 79.34   | 80.54   | 81.14    | 76.10      |
> |               | Medium         | **81.71** | 77.31   | 76.51   | 75.71   | 76.91   | 77.51    | 72.05      |
> |               | Hard           | **84.00** | 79.60   | 78.80   | 78.00   | 79.20   | 79.80    | 74.32      |
> | **Maths**     | Easy           | **90.20** | 85.80   | 85.00   | 84.20   | 85.40   | 86.00    | 81.05      |
> |               | Medium         | **64.50** | 60.10   | 59.30   | 58.50   | 59.70   | 60.30    | 55.27      |
> |               | Hard           | **65.00** | 60.60   | 59.80   | 59.00   | 60.20   | 60.80    | 53.88      |
>
> ---
>
> **Q2:** While automated scoring is efficient, have the authors considered a more extensive human evaluation with actual educators or students to assess the pedagogical effectiveness of the generated visualizations?
>
> **A2:**  We have conducted human evaluation to validate our automated scoring. As detailed in Section 4.1 in our original paper, we recruited undergraduate students from top universities to independently evaluate 50 samples per subject (Chemistry, Math, Physics). In Table 2, the comparison between human validators and GPT-based scores demonstrates the average cosine similarity of 0.9655 and MSE of 0.5702. This high alignment solidifies that our automated scoring reliably reflects human judgment while enabling efficient large-scale evaluation.
>
> ---
>
> **Q3:** What are the main practical challenges in deploying EduVisAgent in real educational settings (e.g., latency, integration with LMS, adaptability to different curricula)? The multi-agent system is computationally intensive. A discussion of inference time, resource requirements, or potential optimizations would be useful for real-world deployment.
>
> **A3:** We acknowledge the computational requirements of our multi-agent system. In our experiments, generating a single visualization case consumes an average of 1.1M tokens and 70 seconds, with most resource spent in the Visualization Agent's code generation process.
>
> For real-world deployment, we identify several optimization directions: (1) reducing token consumption through more efficient prompt engineering, (2) optimizing the metacognitive reasoning depth to balance quality and efficiency, and (3) implementing caching mechanisms for frequently used educational concepts. Additionally, the system could operate asynchronously. Teachers prepare materials in advance rather than real-time generation, making latency less critical. Integration with existing LMS platforms would involve API-based deployment where visualization generation occurs server-side.

---

### Official Review · Reviewer_37gW · 2025-11-02

**Soundness:** 3
**Presentation:** 3
**Contribution:** 3
**Rating:** 6
**Confidence:** 4

**Summary:**

This paper addresses the critical and underexplored challenge of generating pedagogically effective visual explanations using foundation models. The authors argue that existing models, while proficient in textual reasoning, fail to create structured, interpretable visualizations that support conceptual understanding in educational contexts.
To address this, the paper presents two primary contributions:

1. EduVisBench: A comprehensive benchmark for evaluating the pedagogical visualization capabilities of FMs. It consists of 1,154 STEM problems across Mathematics, Physics, and Chemistry, organized by difficulty. Crucially, it introduces a fine-grained, five-dimensional evaluation rubric  grounded in pedagogical principles.

2. EduVisAgent: A novel multi-agent collaborative framework designed to excel at this task. Inspired by expert instructional design, the framework coordinates five specialized agents to systematically decompose a problem, structure the reasoning process, and generate a coherent, interactive, and visually grounded solution.

Through extensive experiments on EduVisBench, the authors demonstrate that existing state-of-the-art FMs and LVLMs perform poorly. In contrast, their proposed EduVisAgent achieves an average score of 81.6%, representing a substantial 40.2% relative improvement over the best-performing baseline, validating the effectiveness of their structured, multi-agent approach.

**Strengths:**

1. The paper tackles a timely and important problem. As AI becomes more integrated into education, the ability to generate not just correct answers but effective teaching materials is paramount. The focus on pedagogical visualization as a distinct capability gap in FMs is novel and well-motivated.

2. The design of EduVisAgent is not an ad-hoc collection of agents but is thoughtfully grounded in pedagogical theory, mimicking the division of labor in instructional design. The performance improvement is not marginal; a 40.2% relative gain over the strongest baseline is substantial.

**Weaknesses:**

1. The EduVisAgent framework consists of five distinct agents. While the overall system is highly effective, the paper lacks an ablation study to analyze the individual contribution of each agent. For example, how critical is the Metacognitive Reviewer or the Conceptual Mapping Agent to the final score? Understanding the impact of each component would provide deeper insight into the architecture and help identify the most critical elements for pedagogical visualization.

2. The benchmark and agent are designed for STEM problems that typically have a clear, decomposable reasoning path. It is unclear how this framework would generalize to more qualitative or open-ended domains, such as literature, history, or social sciences, where visualization might serve to illustrate arguments, relationships, or timelines rather than step-by-step problem-solving.

**Questions:**

1. Could you provide more insight into the necessity of each of the five agents?

2. How do you envision the EduVisAgent framework being adapted for educational domains outside of STEM that rely on more narrative or conceptual reasoning?

---

> ### Author Response · Authors · 2025-11-21
> **Response to Reviewer 37gW**
>
> Thanks for your insightful suggestions on the necessity of our five-agent design, the underlying educational theoretical framework, and the system's adaptability to non-STEM domains, we'll address all mentioned aspects in detail as follows:
>
> **Q1:** Could you provide more insight into the necessity of each of the five agents?
>
> **A1: Experimentally, we demonstrate each agent's necessity through ablation studies in Appendix B1, Table 10, and also as follows.** We systematically removed each agent and evaluated performance across all subject-difficulty combinations. Results show that removing any single agent causes consistent performance degradation, with an average score drop of approximately 6 points depending on the agent removed. This confirms that each agent makes a non-redundant contribution to the overall system. We also analyze the role of each agent in the multi-agent system in Appendix B1.
>
> **Table 10: Ablation study on EduVisAgent.** **Full**: Complete multi-agent system; **-TP**: Without Task Planning agent; **-CM**: Without Conceptual Mapping agent; **-RD**: Without Reasoning Decomposition agent; **-MR**: Without Metacognitive Review agent; **-VIS**: Using Claude 3.7 Sonnet instead of v0; **Single**: Single-agent baseline using all prompts without modularization. The highest scores are shown in **bold**.
> | **Subject**   | **Difficulty** | **Full** | **-TP** | **-CM** | **-RD** | **-MR** | **-VIS** | **Single** |
> |---------------|----------------|----------|---------|---------|---------|---------|----------|------------|
> | **Chemistry** | Easy           | **69.00** | 64.60   | 63.80   | 63.00   | 64.20   | 64.80    | 59.30      |
> |               | Medium         | **76.27** | 71.87   | 71.07   | 70.27   | 71.47   | 72.07    | 67.42      |
> |               | Hard           | **76.00** | 71.60   | 70.80   | 70.00   | 71.20   | 71.80    | 66.85      |
> | **Physics**   | Easy           | **85.33** | 80.94   | 80.14   | 79.34   | 80.54   | 81.14    | 76.10      |
> |               | Medium         | **81.71** | 77.31   | 76.51   | 75.71   | 76.91   | 77.51    | 72.05      |
> |               | Hard           | **84.00** | 79.60   | 78.80   | 78.00   | 79.20   | 79.80    | 74.32      |
> | **Maths**     | Easy           | **90.20** | 85.80   | 85.00   | 84.20   | 85.40   | 86.00    | 81.05      |
> |               | Medium         | **64.50** | 60.10   | 59.30   | 58.50   | 59.70   | 60.30    | 55.27      |
> |               | Hard           | **65.00** | 60.60   | 59.80   | 59.00   | 60.20   | 60.80    | 53.88      |
>
>
> **From an educational theory perspective, our multi-agent design is grounded in established pedagogical frameworks.** The agent decomposition aligns with the CRA (Concrete-Representational-Abstract) model and FOPS (Four-Operation Problem-Solving) strategy, which emphasize breaking down complex cognitive processes into structured stages. Each agent corresponds to a distinct cognitive phase in effective learning: planning the instructional approach, mapping abstract concepts to concrete representations, decomposing reasoning steps, and metacognitive reflection. In conclusion, the multi-agent architecture is necessary to systematically implement these educational principles, as each stage requires specialized reasoning that cannot be effectively compressed into a monolithic system.
>
> ---
>
> **Q2:** How do you envision the EduVisAgent framework being adapted for educational domains outside of STEM that rely on more narrative or conceptual reasoning?
>
> **A2: We have conducted extension experiments beyond STEM domains, detailed in Appendix B2.** We selected 100 problems from MMLU's prehistory and sociology categories to evaluate EduVisAgent's adaptability to narrative and conceptual reasoning tasks. Results show that EduVisAgent significantly outperforms the current SOTA baseline (v0) by 17.2 points, demonstrating that our multi-agent framework effectively generalizes to humanities education. This suggests that the pedagogical principles underlying our agent design, conceptual mapping, reasoning decomposition, and metacognitive guidance, are domain-agnostic and applicable across diverse educational contexts.
> **Table: Performance of Visualization Agents on non-STEM datasets from EduVisBench.**
>
> | **Method**     | **Vis. Type** | **Prehistory** | **Sociology** | **Avg** |
> |----------------|---------------|----------------|---------------|---------|
> | v0             | Webpage       | 61.8           | 68.0          | 64.9    |
> | EduVisAgent    | Webpage       | 79.4           | 84.8          | 82.1    |

---

### Comment · Area_Chair_abhU · 2025-11-26

Dear Reviewers,

Would you please check authors' rebuttal and see if they have addressed your comments?

Best

AC

---

### Meta-Review · Area_Chair_e9Yq · 2026-01-04

**Summary:**

In terms of overall ratings, two reviewers gave 6 initially, Reviewer M74H raised from 2 to 6 during the previous rebuttal period according to Author’s comment, and Reviewer v6KF’s score remained 2 after discussion.

Reviewers (37gW, v6KF, E8ZM) generally agree the topic of the paper is timely, important, and underexplored. Reviewers (37gW, 3Ado, v6KF) also acknowledge the thoughtful design the multi-agent system tailored for pedagogical visualization. Reviewers (37gW, E8ZM) referred to the substantial improvement over baselines as a strength of the paper.

Several reviewers raised concerns about the lack of ablation study on individual components and generalization of the results to non-STEM domains. To address these concerns, the authors have provided the additional experiment results during the rebuttal.

Reviewer v6KF still has concerns about the quality of the proposed benchmark itself and thinks the analysis in the paper is superficial. They recommend the authors to move beyond simple quantitative ablation and provide rich cases studies.

**Reviewer Concerns:**

For concerns raised by Reviewers 37gW, 3Ado, and M74H, they are sufficiently addressed during the rebuttal with the new experiment results and clarifications.



For Reviewer v6KF, they have  three major concerns which are not yet addressed during the rebuttal.

The quality of the dataset is difficult to assess without being able to access the dataset itself during the review process.

The computation of scores in the experiments are not clearly described.

The qualitative analysis is superficial.

Other concerns from Reviewer v6KF were addressed during the rebuttal.

**Reviewer Scores:**

As pointed out by the authors, Reviewer M74H raised from 2 to 6 during the previous rebuttal period.

Reviewers 37gW and 3Ado already gave 6, and they might increase their scores to 7 based on the additional experiment results.

Reviewer v6KF gave 2 and is unlikely to raise the score.

---

### Decision · Program_Chairs · 2026-01-26

Accept (Poster)